



# Upgrading 1D-2D flood models using satellite laser altimetry and multi-mission satellite surface water extent maps

Theerapol Charoensuk[1,2,3], Claudia Katrine Corvenius Lorentzen[1], Anne Beukel Bak[1], Jakob Luchner[2], Christian Tøttrup[2] and Peter Bauer-Gottwein[1]

[1]Department of Environmental and Resource Engineering, Technical University of Denmark, Kgs. Lyngby, 2800, Denmark
[2]DHI A/S, Hørsholm, 2970, Denmark
[3]Hydro-informatics Institute, Bangkok, 10900, Thailand

*Correspondence to*: Theerapol Charoensuk(tcha@dtu.dk)

**Abstract.** Digital elevation models (DEMs) are essential datasets, particularly for flood inundation mapping in one-
10 dimensional (1D) to two-dimensional (2D) flood models. Given the current uncertainties stemming from changes in weather patterns affecting flooding, reducing inaccuracies in flood models is imperative. This study aims to enhance the performance of 1D-2D flood models using satellite Earth observation (EO) data in the lower Chao Phraya (CPY) basin. It introduces two workflows applied to upgrade the 1D-2D flood model: DEM analysis and flood map analysis.

The DEM analysis workflow evaluates 10 DEM products (LDD, JICA, merged LDD-JICA, ASTER GDEM V3, STRMv3,
MERIT, GLO30, FABDEMv1-2, TanDEM-X, and TanDEM-EDEM) using satellite laser altimetry data from the Ice, Cloud, and land Elevation Satellite-2 (ICESat-2) according to standard criteria for DEM selection as input to the flood model. Findings indicate that the merged LDD-JICA and FABDEMv1-2 DEMs exhibit the highest level of accuracy, with root mean square error (RMSE) values of 1.93 and 1.95 m, respectively. The flood map analysis workflow involves comparing flood extent maps derived from multi-mission satellite datasets, and simulated flood maps. This study utilizes surface water
extent (SWE) maps from the WorldWater project, obtained from the Sentinel-1 and Sentinel-2 imaging satellites, and flood maps from the Geo-Informatics and Space Technology Development Agency (GISTDA) in Thailand to verify flood maps produced by the 1D-2D flood model. The results reveal that the flood maps from the 1D-2D flood model tend to overestimate flood extent, with a critical success index (CSI) range of 0.072 – 0.230. Our study demonstrates the potential to enhance the skill of 1D-2D flood models using satellite EO data, thereby improving the reliability of flood inundation
predictions.

## 1 Introduction

Nowadays, flooding is one of the most common disaster issues globally, impacting health, economies, and livelihoods worldwide. Flood models play a crucial role in forecasting floods and assessing flood risks, thereby assisting decision-makers in effective water management, particularly through one-dimensional (1D) - two-dimensional (2D) flood models.
These models simulate various aspects of flooding, including flow, water levels, flood inundation extents, flood depths,





flood maps, and flood duration (DHI Water and Environment, 2019). The Digital Elevation Model (DEM) serves as a primary input parameter for 1D-2D flood models, enabling the accurate simulation of flood overflow from rivers, floodplains, and inundated areas, particularly in flat and low-lying regions. The DEM significantly influences the simulation of flood inundation in both 1D-2D and 2D flood models (Saksena and Merwade, 2015; Shen and Tan, 2020; Wu et al., 2007;

Morrison et al., 2022), urban areas (McClean et al., 2020), coastal areas (Darnell et al., 2008), and flood warning systems (Lamichhane and Sharma, 2018). Ultimately, the reliability of flood inundation predictions relies on the accuracy and detail provided by the DEM, directly impacting the representation of flow geometry characteristics within flood models.

Validating the DEM before integrating it into the 1D-2D flood model is essential. The Ice, Cloud, and Land Elevation Satellite-2 (ICESat-2) is a satellite equipped with a laser altimeter, capable of measuring ice sheet and glacier elevation

change, sea ice freeboard, land elevation, and water elevation (Neumann et al., 2019), providing opportunities for validating DEMs even in remote and hard-to-reach areas worldwide, such as Finland (Wang and Liang, 2023), Spain (Zhu et al., 2022), East Antarctica (Hao et al., 2022), Alaska in the USA (Wang et al., 2019), and the Qinghai-Tibet Plateau in China (Weifeng et al., 2024). Moreover, while an efficient DEM enhances the efficiency of 1D-2D flood simulation, it is important to systematically validate flood maps. Currently, satellite earth observation (EO) data can be utilized for monitoring and

providing surface water extent (SWE) with synthetic-aperture radar (SAR) sensors, such as RADARSAT (Raney et al., 1991), ENVISAT ASAR (Lv et al., 2005), COSMO-SkyMed (Pulvirenti et al., 2014), and TerraSAR-X (Martinis et al., 2013), which is the only way to validate flood inundation maps from flood models over regional scales. The WorldWater project developed a robust and scalable EO solution for inland SWE monitoring, which can be utilized by a large community of stakeholders involved in local water management (Tottrup et al., 2022). The project used free and open optical and SAR

satellite imagery from the Sentinel-1 and Sentinel-2 missions to generate monthly SWE maps over four years, which are accessible from https://worldwater.earth/. The product offers new opportunities for validating modelled flood maps with higher SWE resolution.

While satellite EO provides SWE maps that delineate water bodies and inundated areas, they cannot be directly compared to flood maps from 1D-2D flood models. The output of 1D-2D flood models are riverine flood maps. Additional flood

classification processing is necessary to ensure comparability between SWE maps and the output of a flood model. However, flood type classification using SWE maps poses challenges and difficulties. Many studies focus on classifying flood types based on meteorological condition rather than using SWE maps, such as Nied et al., 2014, Turkington et al., 2016, decision tree using meteorological data (Stein et al., 2019), and Yan et al., 2023. Riverine flood classification specifically involves identifying floods caused by river overflow from SWE maps. Here, we used expanding segmentation labels (ESL) (Van Der

Walt et al., 2014), connected component labeling (CCL) (Rosenfeld and Pfaltz, 1966 and AbuBaker et al., 2007), masking off riverine and permanent water, and morphological image processing (MIP) (Soille, 2003) techniques applied to the SWE maps to separate riverine flood areas from other inundated areas.

This study demonstrates these workflows for the lower Chao Phraya (CPY) River basin in Thailand, i.e. DEM evaluation with ICESat-2 benchmarks, riverine flood classification, and flood evaluation. The evaluation involved 10 DEM products,





including three local DEM products and seven global DEM products. The DEM performance in the lower CPY basin was assessed using statistical methods, including bias (mean error, ME), mean absolute error (MAE), mean square error (MSE), and root mean square error (RMSE), in terms of point, grid, and track-wise comparison. The best result from DEM evaluation was implemented in a 1D-2D flood model for simulating flood inundation. The simulated flood inundation map was evaluated using a satellite EO based flood map, derived from SWE through the riverine flood classification process. The

performance of the flood model was assessed using three statistical metrics: probability of detection (POD), false alarm ratio (FAR), and critical success index (CSI). The methods demonstrated in this paper will enhance the performance of the Chao Phraya operational hydrologic-hydraulic forecasting system maintained at the Hydro-Informatics Institute (HII) in Thailand.

## 2 Study Area

The study area is located in the central part of Thailand, as shown in Figure 1(a). The delta area of the lower CPY River

basin in Thailand forms the study area depicted in Figure 1(c). The size of the study area is approximately 16,643 km$^2$, including about 70% irrigation area and 20% urban area. The topography of the study area is characterized by a flat terrain, predominantly consisting of a low-lying alluvial floodplain. The northern part of the study area is a mountainous region with four main rivers: the Ping, Wang, Yom, and Nan rivers. These rivers converge to form the CPY river, which then flows into the study area. The eastern and western parts of the study area are connected to the Bang Pakong River and the Mae Klong

basin, respectively. The southern part of the study area borders the Gulf of Thailand.

The study area is located in a tropical climate and is influenced by northeast and southwest monsoons. The northeast monsoon brings cool and dry air from November to February, while the southwest monsoon brings humid air from May to October. The precipitation is approximately 1,100 mm during the rainy season and 170 mm during the dry season. The flooding in the study area is caused by the main rivers and their tributaries. The tributaries of the CPY river include Tha-

85 Chin, Noi, and Lopburi. Flooding problems are more severe along the main course of the CPY river compared to others. Nevertheless, flooding mechanisms are complicated, arising from the combined effects of extreme precipitation, river overflows, insufficient river conveyance, land-use change, and sea-level rise. This results in frequent flooding, as shown in Figure 3(c).

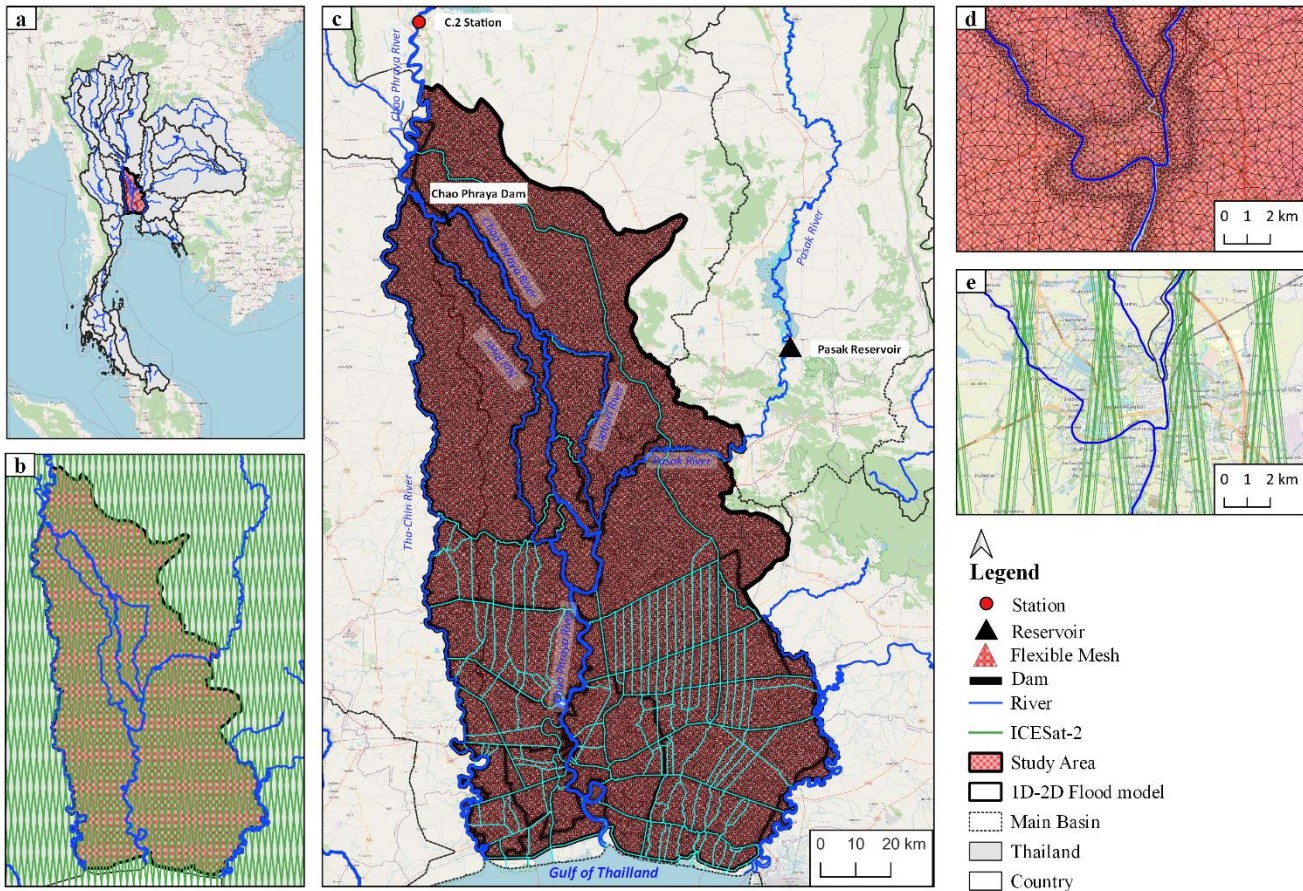

**Figure 1: (a) the location of the study area, (b) the ICESat-2 orbit, (c) the study area/1D-2D flood model, (d) the flexible mesh in the flood model, and (e) ICESat-2 beam pairs. © OpenStreetMap contributors 2015. Distributed under the Open Data Commons Open Database License (ODbL) v1.0.**

## 3 Materials

### 3.1 1D-2D Flood modelling

In this study, we used the flood model from the decision support system for flood forecasting and water management in the CPY River basin, developed in collaboration with HII and DHI A/S since 2012 (Sisomphon et al., 2013) and updated with new information in 2016 (Charoensuk et al., 2018). The decision support system for flood forecasting and water management in the CPY basin continues to operate, supporting the Thai Government in managing flood risk and providing real-time flood forecasts.

The flood model used the MIKE FLOOD software developed by DHI A/S. A MIKE FLOOD model (DHI Water and Environment, 2019 consists of coupled one-dimensional (1D) and two-dimensional (2D) models, namely MIKE11 and



MIKE21, respectively. The 1D hydraulic model (MIKE11) simulates unsteady flow in river networks solving the Saint-Venant equations with an implicit finite difference solver (DHI Water and Environment, 2021). The main branches of MIKE11 include the Chao Phraya, Tha-Chin, Lopburi, Noi and Pasak rivers. Cross-sections, rainfall-runoff, boundary
conditions, hydrodynamic parameters, and control structures were implemented in MIKE11. The MIKE21 model is an overland flow model utilizing 2D shallow water equations (Danish Hydraulic Insitute, 2016). MIKE21 employs a 2D flexible mesh based on the digital elevation model (DEM) to assess flood depth and its propagation. The river network in MIKE11 is dynamically linked to floodplain bathymetry through lateral links. The lateral links connect the river to the floodplain along its length using the cell-to-cell method, allowing water to overflow to the floodplain in the MIKE21
overland flood model. The lateral link connection uses the weir equation to calculate overflow in MIKE FLOOD (DHI Water and Environment, 2019).

The 1D-2D flood model, documented in Hydro-Informatics Institute, 2017, establishes the following boundary conditions: upstream boundary forcing with discharge from C.2 station and releases from the Pasak Reservoir from the Royal Irrigation Department (RID) in the CPY and Pasak rivers, respectively. Meanwhile, the downstream boundary connects to the Gulf of
115 Thailand using sea level measurements from the Hydrographics Department, Royal Thai Navy (NAVY), as illustrated in Figure 1(c). The MIKE11 model was calibrated using water level observations presented in Charoensuk et al., 2024. MIKE21 utilized a flexible mesh to simulate overland flow, as illustrated in Figure 1(d), and MIKE FLOOD was calibrated against flood maps and satellite data from 2011, as detailed by Charoensuk et al., 2018.

### 3.2 Geoid Models

To measure elevations around the Earth, a vertical reference is needed, with mean sea level chosen as the reference. The geoid is the level (equipotential) surface of the Earth's gravity field that best coincides with mean sea level. This surface connects the oceans and extends through the continents. The geoid serves as the reference surface for levelled heights, commonly expressed as 'heights above sea level'. In order to compare heights from different data sources, all data has to be re-referenced to the same geoid model. A geoid model is a spatial representation of geoid height, encompassing both global
and local scales. This study has collected three geoid models, summarized in Table 1 and shown in Figure A 2. Thailand has its own local geoid model. The latest one, TGM2017, was released in 2018. This geoid is based on new gravity measurements taken around Thailand and has been shown to better match the expected geoid heights than the EGM2008 model (Dumrongchai et al., 2021). TGM2017 provides the best fit for Thailand, it was chosen as the primary geoid model, and all heights were re-referenced to TGM2017.

**Table 1: Geoid model datasets**

| Geoid model | Scale | Download | References |
|---|---|---|---|
| EGM96: The Earth Gravitational Model 1996 | Global | https://earth-info.nga.mil/ | Lemoine et al., 1998 |
| EGM2008: The Earth Gravitational Model 2008 | Global | https://earth-info.nga.mil/ | Pavlis et al., 2012 |
| TGM2017: Thailand geoid model 2017 | Local | On request | Dumrongchai et al., 2021 |



## 3.3 Digital Elevation Models (DEM)

A digital elevation model (DEM) is a quantitative representation of the Earth's surface elevation. The term "DEM" encompasses both digital terrain models (DTM) and digital surface models (DSM). A DSM maps the heights of all features on the surface, such as vegetation and buildings, while a DTM only represents the actual height of the terrain ("bare earth"). Multiple digital elevation models are available, local DEMs are often preferred due to their higher spatial resolution and vertical accuracy (McClean et al., 2020). In this study, we have collected 10 DEM products, as shown in Figure 2, and they are summarized in Table 2. The three local DEM products were obtained from the Thai agency, namely LDD DEM, JICA DEM, and merged LDD-JICA DEM. Additionally, seven global DEMs were collected, including ASTEM GDEM V3, SRTMv3 DEM, MERIT DEM, FABDEMv1-2 DEM, GLO30 DEM, TanDEM-X, and TanDEM-EDEM.

**Table 2: Digital Elevation Model (DEM)**

| Dem Product | Spatial resolution | Data Collection (Year) | Datum Reference | Type | Scale | Acquisition technique |
|---|---|---|---|---|---|---|
| LDD DEM | 5 m | 2004 | EGM96 geoid | DSM | Local | Aerial stereo photo |
| JICA DEM | 2 m | 2012 | EGM2008 geoid | DTM | Local | Airborne LiDAR |
| merged LDD-JICA DEM | 2 m | - | TGM2017 geoid | Between DSM and DTM | Local | Fusion of multisource data |
| ASTEM GDEM V3 | 1 arcsecond (~30 m) | 2000-2010 | EGM96 geoid | DSM | Global | Satellite stereo images |
| SRTM DEM V3 | 1 arcsecond (~30 m) | 2000 | EGM96 geoid | DSM | Global | SAR Interferometry |
| MERIT DEM | 3 arcseconds (~90 m) | 2000 | EGM96 geoid | DSM | Global | Fusion of multisource data |
| GLO30 DEM | 1 arcsecond (~30 m) | 2011-2015 | EGM2008 geoid | DSM | Global | Fusion of multisource data |
| FABDEM v1-2 | 1 arcsecond (~30 m) | 2011-2015 | EGM2008 geoid | Base on DSM remove building and forest | Global | Fusion of multisource data |
| TanDEM-X DEM | 0.4 arcsecond (~12 m) | 2011-2015 | WGS84 ellipsoidal height | DSM | Global | SAR Interferometry |
| TanDEM-X EDEM | 1 arcsecibd (~30 m) | 2011-2015 | WGS84 ellipsoidal height | DSM | Global | Fusion of multisource data |


**Figure 2: ICESat-2 ATL08 and DEM products**



### 3.3.1 LDD DEM

The LDD DEM data is supplied by the Land Development Department of Thailand (LDD) in a grid format, with a resolution of 5x5 meters. This DEM was generated using photogrammetry using aerial stereo photo pairs with known scales (Paengwangthong and Sarapirome, 2012). This approach involves deducing distances between points from photos and determining object heights by identifying stereoscopic parallax from multiple pictures and rectifying with ground control points (GCPs) (Sholarin and Awange, 2015). Subsequently, orthorectification and interpolation are used to generate a DEM

and mask off buildings and vegetation. Because buildings and vegetation are removed, the LDD DEM approximates a DTM (Sholarin and Awange, 2015).

### 3.3.2 JICA DEM

The JICA DEM was produced through a collaborative effort between the Royal Irrigation Department (RID) and the Japan International Cooperation Agency (JICA) at a resolution of 2x2 meters (Japan International Cooperation Agency (JICA),

2012).The JICA DEM was generated using Airborne Laser Scanning techniques with the LiDAR (Light Detection And Ranging) aerial technology. The LiDAR aerial survey employs a pulse laser to measure distances between the target and sensor, and it is applied on a large scale. The distance from the vehicle to the surface can be determined based on the travel time of the laser pulse (Argall and Sica, 2003). The JICA DEM was processed into a DTM filtering out features such as transportation facilities, buildings, and vegetation from the original data, as described in Japan International Cooperation

Agency (JICA), 2012.

### 3.3.3 Merged LDD-JICA DEM

The merged LDD-JICA DEM was generated by integrating the LDD DEM and JICA DEM as described by Charoensuk et al., 2018. The JICA DEM served as the primary dataset, while the LDD DEM was utilized in areas with gaps within the 1D-2D Flood modeling boundary. To incorporate the LDD DEM into the merged LDD-JICA DEM within data gaps, we applied

bias correction. The native LDD DEM and JICA DEM datasets were not referenced to the same vertical datum. The processing of the merged LDD-JICA DEM consists of two primary steps (Figure A 3): 1) re-referencing both LDD DEM and JICA DEM to the TGM2017 reference, and 2) calculation of the correlation coefficient between the JICA and LDD DEM for 1000 random points, using linear regression to correct the bias in the LDD DEM, as shown in Figure A 4. Following this, the JICA DEM and LDD DEM are combined to create the merged LDD-JICA DEM using linear regression.

The resulting combined merged LDD-JICA DEM has a resolution of 2x2 meters.

### 3.3.4 ASTER GDEM3

The Advanced Spaceborne Thermal Emission and Reflection Radiometer (ASTER GDEM3), serving as a global DEM, was developed by the Ministry of Economy, Trade, and Industry (METI) of Japan in collaboration with The United States



National Aeronautics and Space Administration (NASA) and was published on 2019. The footprint of ASTER GDEM spans

latitudes from 83°N to 83°S. The study area utilized ASTER GDEM3 (Abrams et al., 2020), which can be downloaded from the associated website: https://gdemdl.aster.jspacesystems.or.jp/. More information is shown as Table 2.

### 3.3.5 SRTMv3 DEM

The Shuttle Radar Topography Mission (SRTM) DEM, developed by NASA, was a collaborative effort involving the National Geospatial-Intelligence Agency (NGA) and the German and Italian Space Agencies. It was part of an international

project aimed at acquiring radar data, which were used to create the first near-global set of land elevations (Werner, 2001). The DEM was launched in 2000 (Farr et al., 2007), and many improvements have been made since then. The SRTMv3 DEM, the latest version, was used for the study area and can be downloaded from the associated website: https://search.earthdata.nasa.gov/search.

### 3.3.6 MERIT DEM

The Multi-Error-Removed Improved-Terrain (MERIT) DEM, developed by Yamazaki et al., 2017. MERIT DEM improves upon previous DEMs by systematically removing various error components such as absolute bias, stripe noise, speckle noise, and tree height biasn from SRTM3 DEM (Farr et al., 2007) , AW3D-30 m DEM (Tadono et al., 2015) and gap-filling with the Viewfinder Panoramas (VFP) DEM (http://viewfinderpanoramas.org/dem3.html). The MERIT DEM is a DSM with resolution of 3 arc seconds. It was utilized for the study area and is available for download from the dedicated website:

http://hydro.iis.utokyo.ac.jp/~yamadai/MERIT_DEM/index.html/.

### 3.3.7 GLO30 DEM

The Copernicus DEM, published in 2019 by the European Space Agency (ESA) (AIRBUS, 2020), represents an upgraded iteration of the WorldDEM. The backbone of the Copernicus WorldDEM is the TanDEM-X mission data, yet void filling techniques and integration of other data sources are used to enhance data completeness and accuracy. The Copernicus DEM

is provided in three different DSM instances: EEA-10, GLO-30, and GLO-90. For this study, GLO-30 was utilized, offering 1 arc-second resolution. It can be downloaded from the dedicated website: https://spacedata.copernicus.eu/de/collections/copernicus-digital-elevation-model.

### 3.3.8 FABDEMv1-2

Forest And Building removed Copernicus Digital Elevation Model (FABDEM) was developed in collaboration between

200 Bristol-based flood modelling company Fathom and the University of Bristol FloodLab. The FABDEM V1-0, launched in 2021 (Laurence Hawker, 2021), is derived from the Copernicus GLO-30 (AIRBUS, 2020) DSM. FABDEM V1-2, released in 2023 (Hawker et al., 2023), has a 1 arc-second resolution and is based on a DSM which removes buildings and vegetation.



This dataset was employed for the study area and is available for download from https://data.bris.ac.uk/data/dataset/s5hqmjcdj8yo2ibzi9b4-ew3sn.

### 3.3.9 TanDEM-X DEM

TanDEM-X (TerraSAR-X add-on for Digital Elevation Measurement) is an innovative space borne-radar interferometer based on two TerraSAR-X radar satellites flying in close formation (Krieger et al., 2007). The TanDEM-X mission represents a collaborative effort between the German Aerospace Center (DLR) and AIRBUS (Wessel, 2016), with the aim of generating a globally consistent DEM. TanDEM-X, launched in 2016, is a DSM with resolutions of 0.4, 1, and 3 arcseconds. The 3-arcsecond TanDEM-X product is readily accessible and can be downloaded directly from https://geoservice.dlr.de/data-assets/ju28hc7pui09.html. However, the 0.4 and 1 arcsecond products are available from DLR upon request. It is important to note that the TanDEM-X product has not undergone full processing to eliminate artifacts, outliers, noisy regions, and data gaps. As a result, its adoption in flood modeling has been limited (McClean et al., 2020). In this study, we employed TanDEM-X with a 0.4 arcsecond resolution for our flood modeling purposes.

### 3.3.10 TanDEM-EDEM

The TanDEM-X Edited Digital Elevation model (TanDEM-EDM) is an edited version of the TanDEM-X Global with a 1-arcsec (~30 m) pixel resolutionreleased in 2023 (Wessel, 2016). The main update in TanDEM-EDEM version 1 includes filling gaps with suitable alternative DEM data and improving representation of water bodies. The TanDEM-EDEM dataset, which is a DSM, was utilized for the study area and is readily available for download from: https://download.geoservice.dlr.de/TDM30_EDEM/. It has a resolution of 30 m.

### 3.4 ICESat-2 satellite laser altimetry

Ice, Cloud, and Land Elevation Satellite-2 (ICESat-2) is a laser altimetry satellite launched by the National Aeronautics and Space Administration (NASA) in 2018. As the follow-on satellite of ICESat, ICESat-2 continues elevation measurements of ice sheets, glaciers, sea ice, and various other land features with a 91-day exact repeat orbit. ICESat-2 carries the Advanced Topographic Laser Altimeter System (ATLAS), which works by transmitting 10,000 laser pulses per second using laser light of 532 nm (Neumann et al., 2019). The pulse rate enables the satellite to capture a measurement every 70 cm along the ground track. The pulse divides into six beams, organized into three pairs. Each pair comprises one right-side beam and one left-side beam, striking the Earth at a 90 m distance from each other. The distance between each pair is 3.3 km, as depicted in Figure 1(E).

The National Snow and Ice Data Center (NSIDC) portal has developed various products that incorporate photon travel times and locations determined using the built-in GPS from the ICESat-2 satellite. This mission generates 21 products, as detailed on their website: https://nsidc.org/data/icesat-2/products. The two data products used in this study are ATL03 and ATL08, as





summarized in Table 3, and the groundtrack pattern of ICESat-2 in this study area is shown in Figure 1(b). The ICESat-2 data were obtained from the NSIDC website via their data access tool (https://nsidc.org/data/data-access-tool).

**Table 3: ICESat-2 product**

| ICESat-2 product | | Data Collection (Year) | Datum Reference |
|---|---|---|---|
| ATL03 | Global Geolocated Photon Data (DSM) | 2018 - 2022 | WGS84 ellipsoid |
| ATL08 | Land/Water/Vegetation Elevation (DSM) | 2018 - 2022 | WGS84 ellipsoid |

### 3.4.1 ATL03

The ATL03 data product from ICESat-2 plays a crucial role as an intermediary between lower and higher-level products. It provides time, latitude, longitude, and height information for each track photon. This comprehensive source of photon data facilitates subsequent analyses and enables the generation of surface-specific products, such as land ice height and sea ice

freeboard. ATL03 includes surface masks and photon event classifications and applies geophysical corrections to enhance accuracy. Additionally, it supplies spacecraft and instrument information required by higher-level data products. Its role is crucial in facilitating streamlined data processing and serving as a unified source for further scientific investigations (Tom Neumann et al., 2021).

### 3.4.2 ATL08

The ATL08 product is derived from the ATL03 product. The ATL08 product offers estimates of terrain heights, canopy heights, canopy cover, and other descriptive parameters at fine spatial scales in the along-track direction. A fixed segment size of 100 m was chosen to provide continuity of data parameters on the ATL08 data product. Height estimates from ATL08 can be compared with other geodetic data and serve as input for higher-level products like ATL13 (inland water-related heights) and ATL18 (terrain and canopy feature maps) (Neuenschwander et al., 2022). In this study, we used ATL08

product from ICESat-2 as the benchmark.

### 3.5 Flood Map/Surface Water Extent (SWE) dataset

In this study, SWE and flood maps were collected from two sources: surface water extent (SWE) data from the WorldWater project (https://worldwater.earth/), funded by the European Space Agency (ESA) and the Geo-Informatics and Space Technology Development Agency (Public organization) (GISTDA) in Thailand. The flood map datasets are summarized in

Table 4 and presented in Figure 3.

### 3.5.1 WorldWater Surface Water Extent (SWE)

We used SWE products from the WorldWater project, and using data from the Sentinel-1 and Sentinel-2 imaging satellites, both integral parts of the ESA Copernicus program. The Sentinel-1 satellite, launched in 2014, is equipped with a SAR constellation consisting of two polar-orbiting satellites, with objectives on land and ocean monitoring. Sentinel-1 comprises





a C-band SAR sensor with a 10-meter spatial resolution, operating at an orbiting altitude (Torres et al., 2012). The Sentinel-2 satellites consist of two satellites, namely Sentinel-2A and Sentinel-2B, launched in 2015 and 2017, respectively. The dual-satellite system operates in coordination with a 180° phase difference in the sun-synchronous orbit, supporting both land and ocean monitoring (European Space Agency (ESA), 2015). The WorldWater SWE mapping algorithm utilized Sentinel-1 and Sentinel-2 data from 2017 to 2021 to develop a SWE dataset. The details of Sentinel-1 and Sentinel-2 coverage across the

study area are depicted in Table A 2. This algorithm utilizes a fusion approach (Tottrup et al., 2022), combining optical and radar observations, to provide a more robust delineation of water surfaces. The SWE products provide information on water occurrence, monthly water presence, water seasonality, maximum and minimum water extent, all accessible on the website: https://swdap.worldwater.earth/. The monthly water presence of the Worldwater SWE in November 2017 is illustrated in Figure 3(c). It is important to note that the WorldWater SWE dataset uses a median composite of all Sentinel-1 and Sentinel-

2 acquisitions within a given month to predict monthly surface water presence. Consequently, it does not necessarily reflect the maximum extent of flooding within that month.

### 3.5.2 GISTDA Flood map

GISTDA is a Thai space agency and space research organization that utilizes satellites such as Cosmo-SkyMed, KOMPSAT, LANDSAT-5, RADARSAT-2, and THAICHOTE (Channumsin et al., 2020) to conduct research and development. GISTDA

receives observations of the Earth through the use of Synthetic Aperture Radar (SAR) and optical sensor satellites (Nithirochananont et al., 2010). SAR satellite information is derived from two constellations: RADARSAT and the Advanced Land Observing Satellite (ALOS). RADARSAT comprises two SAR satellites, while ALOS integrates a SAR satellite with an optical satellite. Both RADARSAT and ALOS possess SAR data processing systems. In flooded areas, the Earth's surface appears smooth in the wavelength of the SAR. This smooth surface causes microwaves to reflect in a

specular way, resulting in low backscatter values. This characteristic allows for real-time flood imaging and identification. The SAR data undergoes processing, and image quality enhancement while eliminating any noise present in the data products (Auynirundronkool et al., 2012).

To generate flood maps from satellite data, GISTDA employed several analysis methods, including supervised classification, visual analysis, and thresholding, which were combined with field images. Subsequently, GISTDA used the boundaries of

285 natural and permanent water sources from the existing database and removed these areas from the flood map. Since 2005, GISTDA has annually published nowcasting flood maps and flood occurrence maps on https://flood.gistda.or.th/, which were utilized in this study. The GISTDA flood occurrence map is shown in Figure 3(b).

From 2014 to 2023, HII analysed flood frequency maps from GISTDA. The assessment focused on the frequency of flood occurrences, which were categorized into three levels: low, medium, and high-risk flood frequency. Low-risk flood

frequency is defined as 1-3 occurrences within the 10-year span, medium risk as 4-7 occurrences, and high risk as 8-10 occurrences, as depicted Figure 3(a).





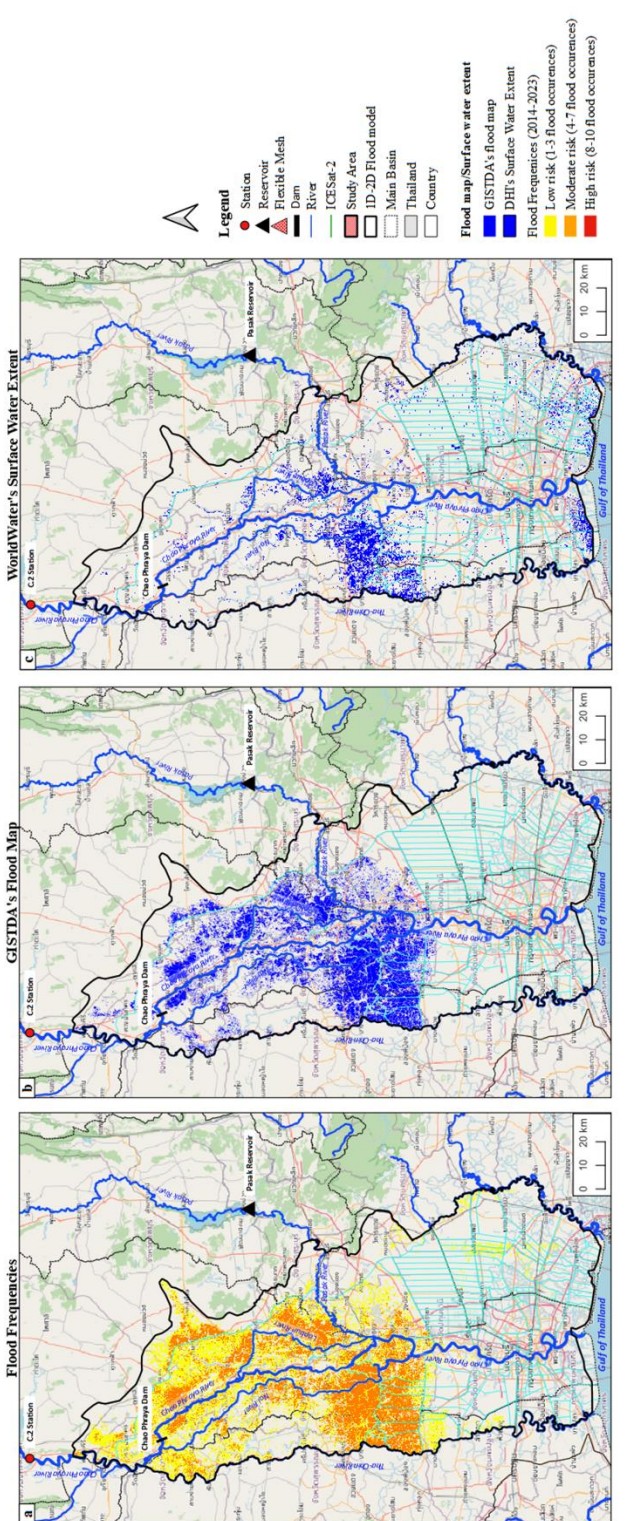

**Figure 3:** The flood map/surface water extent in study area, presenting (a) flood frequency from HII, (b) GISTDA's flood map in Nov 2017, and (c) WorldWater's SWE in Nov 2017. © OpenStreetMap contributors 2015. Distributed under the Open Data Commons Open Database License (ODbL) v1.0.





**Table 4: Flood map datasets**

| Product | Resolution (m.) | Period | Frequency | Type file | Download |
|---|---|---|---|---|---|
| GISTDA's flood map | - | 2005-2021 | On request, satellite track, and annual | Shape file | https://flood.gistda.or.th/ |
| WorldWater's surface water extent | 10 | 2017-2021 | Monthly and annual | Raster file | https://swdap.worldwater.earth/ |

## 4 Methodology

The workflow used in this study, illustrated in Figure 4, comprises two primary components. The first component, namely

DEM analysis, focuses on evaluating the DEMs (Sect. 3.3) with the ICESat-2 benchmark (Sect. 3.4). The best DEM identified in the DEM analysis is then used as input to the flood map analysis. The flood map analysis, focuses on evaluating flood maps generated by the 1D-2D flood model (Sect. 3.1) against WorldWater SWE and GISTDA flood map (Sect. 3.5).

### 4.1 DEM Analysis

The primary objective is to assess the accuracy and reliability of the DEMs by comparing them with elevation data obtained

from ICESat-2 using statistical methods. In the study area, ICESat-2 ATL08 data were primarily used for evaluation, while ICESat-2 ATL03 data were employed in complex terrain. Figure 4(a) illustrates the workflow involving processing and re-referencing steps. Subsequently, the evaluation of DEMs and ICESat-2 is conducted using statistical methods.

### 4.1.1 ICESat-2 ATL08 Data Processing

ATL08 provides estimates of terrain height, canopy height, and canopy cover at fine spatial scales in the along-track

direction. For each parameter, terrain surface elevation and canopy heights were provided at a fixed along-track segment size of 100 meters (Neuenschwander et al., 2022). The AT08 dataset comprises a total of 18 land parameters, such as Mean terrain height for segment (h_te_mean), Mode of terrain height for segment (h_te_mode), Number of ground photons in segment (n_te_photins), slope of terrain within segment (terrain_slope), Best fit terrain elevation at the 100 m segment mid-point location (h_te_best_fit), and others. We processed the ATL08 dataset, extracting the latitude and longitude of the

photon signals along with the photon heights above the WGS84 ellipsoid. The terrain elevation parameter used for evaluation was h_te_best_fit.





**Figure 4: Overall methodology (a) Component 1: DEM analysis and (b) Component 2: Flood map analysis**



### 4.1.2 Vertical Datum Reference Processing

To evaluate the DEMs with the ICESat-2 benchmark, it is necessary to use the same vertical datum reference. Vertical datum reference processing was employed to standardize the datum reference. In this study, the vertical datum reference was TGM2017, using Eq. (1) to establish accurate measurements of vertical elevation

$$H = h - N \tag{1}$$

Where $H$ is ortometric height, $h$ is ellipsoid height, and $N$ is geoid height.

$$H_{DEM\,ref\,TGM2017} = h_{DEM} + N_{DEM} - N_{TGM2017} \tag{2}$$

Where $H_{DEM\,ref\,TGM2017}$ is the DEM referenced to TGM2017, $h_{DEM}$ represents the original DEM, $N_{DEM}$ is the geoid reference of the original DEM, $N_{TGM2017}$ is TGM2017 geoid model.

To obtain DEMs referenced to TGM2017, EGM96 and EGM2008 height corrections were added to the DEM heights, followed by subtracting the TGM2017 geoid corrections, as shown in Eq. (2). The geoid model datasets are shown in Sect. 3.2 for reference. For ICESat-2 elevations referenced to TGM2017, the TGM2017 correction was subtracted from the ICESat-2 elevation data.

### 4.1.3 Evaluation of DEMs using ICESat-2 ATL08 Benchmark

The DEM products were estimated and evaluated using statistical methods, including bias (mean error, ME), mean absolute error (MAE), mean square error (MSE), and root mean square error (RMSE). The overall purpose of implementing these statistical methods is to evaluate the paired ICESat-2 ATL08 data with the 10 DEM products covering the study area.

$$ME = \frac{1}{n}\sum_{i=1}^{n}(Y_i - \hat{Y}_i) \tag{3}$$

$$MAE = \frac{1}{n}\sum_{i=1}^{n}|Y_i - \hat{Y}_i| \tag{4}$$

$$MSE = \frac{1}{n}\sum_{i=1}^{n}(Y_i - \hat{Y}_i)^2 \tag{5}$$

$$RMSE = \sqrt{\sum_{i=1}^{n}\frac{(Y_i - \hat{Y}_i)^2}{n}} \tag{6}$$

Where $\hat{Y}i$ represents ICESat-2 ATL08 elevation, $Yi$ denotes the elevation for each DEM (i.e., LDD DEM, JICA, merged LDD-JICA DEM, ASTEM GDEM V3, SRTM DEM, MERIT DEM, FABDEM v1-2 DEM, GLO30 DEM, TanDEM-X, and TanDEM-EDEM), and $n$ is the number of observations.

We conducted three types of comparisons as follows:

**Point comparison**

Point comparison was performed for every segment of the ICESat-2 ATL08 pass over the study area. This approach aimed to provide a quantitative overview of the quality and identify potential discrepancies among the ten DEMs in comparison to





ICESat-2 ATL08 data (Weifeng et al., 2024), using statistical methods. A total of 954,800 elevation points were extracted
from the study area for point-to-point comparison.

**Grid comparison**

The grid comparison was conducted using a regular square grid over the study area. This comparison provides an overview
of the spatial variation of the quality of the DEMs in comparison to ICESat-2 ATL08 benchmark. In this study, we employed
a 5-km resolution for grid comparison, which involved calculating statistical measures for every segment within each grid
cell and displaying the evaluation spatially on a map.

**Track-wise comparison**

The track-wise comparison was conducted using tracks of ICESat-2 over the study area. The distance between the ICESat-2
points was calculated using UTM x and y coordinates, as shown in Eq.(7). The track-wise comparison represents an overall
elevation profile comparison between DEMs and ICESat-2 ATL08 data over the study area.

$distance = \sqrt{(x_0 - x_i)^2 + (y_0 - y_i)^2}$                      (7)

Where $x$ represents the x coordinates, and $y$ denotes the y coordinates.

## 4.2 Flood Map Analysis

The purpose of flood map analysis is to evaluate the performance of simulated flood maps from the 1D-2D flood model
using various DEM products selected from the first component in comparison to the WorldWater SWE and GISTDA flood
map. This comparative analysis aims to assess the accuracy and effectiveness of the improved flood simulation model.

### 4.2.1 1D-2D Flood Modelling Setup

The setup of the 1D-2D flood model mirrored the original model, retaining the same parameters with only the DEM being
modified to generate the flood map. The DEM products were selected based on the evaluation of DEMs against the ICESat-2
ATL08 benchmark. Flood maps in the lower CPY basin were simulated using the 1D-2D flood model for the years 2017,
and 2021. The flood map simulation results from the 1D-2D flood model present flood extents that occurred during the
simulation period and at each daily time step (DHI, 2018). In this study, we employ simulated flood maps generated from a
1D-2D flood model using the merged LDD-JICA DEM and FABDEMv1-2 DEM products and compare them with
WorldWater SWE and GISTDA flood maps.

### 4.2.2 Flood Classification Processing

The flood map and SWE dataset used for evaluation in this study (Sect. 3.5) had different resolutions, formats, and flood
map definitions. To effectively assess the simulated flood map from 1D-2D flood model, we compared it to the WorldWater
SWE and GISTDA flood map. However, it is crucial to employ the same resolution, format, and flood definition. Common
types of flooding include flood irrigation, pluvial flash floods, coastal floods, and riverine floods. The 1D-2D flood model





only simulates riverine floods, caused by high water levels in the rivers, eventually overflowing onto the neighboring land
due to high river discharge over an extended period. In order to compare the simulated flood map to the satellite EO
products, we first have to extract riverine flooding patterns from the surface water extent maps provided by satellite EO. This
is done in the following steps:

**Permanent water processing**

Permanent water bodies should be removed from the satellite EO SWE maps prior to comparison. The GISTDA datasets
does not include permanent water bodies. The WorldWater product includes permanent water bodies, which must be
removed prior to comparison with simulated flood maps. We use relative water frequency (Yamazaki et al., 2015), which
measures the occurrence of surface water within a defined time period. The relative water frequency $fr$ of pixel was defined
by Eq.(8) and shown as Figure A 5(a).

$$f_r(t) = \frac{f_a(t)}{f_v(t)} \tag{8}$$

Where $fa$ depicts the frequency of surface water detections during a certain time period for each pixel, and $fv$ represents the
frequency of valid observations during the same period for each pixel.

The relative water frequency ranges between 0.0 to 1.0. The permanent water designation indicates that there was observed
water coverage in every single observation of the considered time period, which corresponds to a relative water frequency of
1.0 (Martinis et al., 2022). In many cases, lower thresholds of 0.9, 0.7, and 0.5 were applied (Rao et al., 2018; Yamazaki et
al., 2015). The permanent water map for each threshold is illustrated in Figure A 5. In this study, the threshold for relative
water frequency is set to 0.7, indicating that a pixel is considered permanent water if it is present in 70% or more of the valid
observations over the specified time period. The output of the permanent water processing is utilized in riverine flood
classification processing to remove permanent water from the WorldWater SWE.

**Riverine flood classification processing**

The WorldWater and GISTDA datasets contain both riverine floods and other inundated areas caused, for instance, by
irrigation or pluvial floods. In order to separate riverine floods in the satellite EO flood maps, we used the following method
Figure A 6:

▪ Expand the wet area from WorldWater and GISTDA by 200 meters using expand segmentation labels (ESL) without
overlap (Van Der Walt et al., 2014). The ESL method merges labels in a label image based on the distances between
405 each pixel. Labels that are close by will be merged.

▪ Subsequently, label each pixel using connected component labeling (CCL) (Rosenfeld and Pfaltz, 1966 and AbuBaker et
al., 2007). The CCL method is employed to detect connected regions in binary digital image. The assumption of riverine
flood identification is based on the presence of wet connected pixels originating from the river. These are then masked
off using ESL, and the riverine flood label is selected.

▪ Subsequently, the SWE undergoes morphological image processing (MIP) using a closing algorithm (Van Der Walt et
al., 2014). The structuring element, footprint, passed to the closing algorithm is a boolean array describing the





neighborhood. We used a disk to create a circular structuring element with a radius of 2, implemented as the footprint. The output provides riverine flood maps, namely WorldWater and GISTDA flood map, for evaluation with other flood map products.

### 4.2.3 Flood map evaluation methods

This study evaluates the flood map of the lower CPY River basin using the contingency table ("Glossary of Terms," 1998), comparing flood maps from two different dimensions, as shown in Table 5. We evaluated the flood maps produced by the 1D-2D flood model by comparing them with the monthly surface water presence maps from WorldWater and GISTDA for the years 2017 and 2021. We mainly use probability of detection (POD), false alarm ratio (FAR), and critical success index (CSI) (Forecast, 1995) to perform the evaluation. These statistics are based on the number of grid cells or pixels in the study area is defined as:

$$POD = \frac{Hit}{Hit+Miss} \tag{9}$$

$$FAR = \frac{False\ alarm}{Hit+False\ alarms} \tag{10}$$

$$CSI = \frac{Hit}{Hit+False\ alarms+Miss} \tag{11}$$

Where, *Hit* represents the number of correctly detected flooded pixels from two different dimensions. *True negative* donates the number of correctly detected non-flooded or dry areas from two different dimensions. *Miss* indicates the number of floods from dimension 1 that are not detected by dimension 2, while *False Alarm* represents the number of floods from dimension 2 which did not occur floods in dimension 1. A perfect score for both POD and CSI is 1, while a value of 0 represents the best score for FAR.

**Table 5: Contingency table**

| | | Observation flood map | |
|---|---|---|---|
| | | **Flood** | **Unflood** |
| **Model flood map** | **Flood** | Hits | False alarms |
| | **Unflood** | Misses | True negative |

## 5 Result

### 5.1 1D-2D Flood model calibration results

The 1D river model was calibrated using in-situ water surface elevation data for the period 2012 to 2013. The calibration results of the main river in the study area are presented in Charoensuk et al., 2024. The overall performance during the calibration period is generally satisfactory for all main rivers, with an average $R^2$ of 0.96, RMSE of 0.30 m, and NSE of 0.90. The 1D-2D flood model has been calibrated for extreme floods in 2011, as presented in Charoensuk et al., 2018. Normally, flooding in Thailand is influenced by meteorological conditions, river conveyance, and sea level rise. However, the primary cause of the 2011 flood was dike breaching along the Chao Phraya River, resulting in uncontrollable flood inundation. The



simulated flood, when compared with GISTDA's flood map, satisfactorily corresponds to flood depth, flood propagation
direction, and duration.

**5.2 Results of DEMs evaluation against the ICESat-2 ATL08 benchmark**

**5.2.1 Point comparison evaluation results**

Figure 5 illustrates point comparison between the statistical metrics of 10 DEM products against ICESat-2 ATL08
benchmark. As depicted in the Figure 5(a), the average ME of the local DEM products was -0.88 m, whereas the average
ME of global DEM products was +1.62 m. The results indicate that local DEM products tend to underestimate, while global
DEM products tend to overestimate when compared against ICESat-2 ATL08 benchmark. This tendency be attributed to the
algorithms described in Sect. 3.3, which remove buildings and vegetation from the local DEM products. Moreover, the local
DEM products have a finer grid resolution compared to the global DEM products.  The average performance statistics of the
local DEM and global DEM were 1.25 and 2.17 m for MAE, 4.23 m and 13.52 m for MSE, and 2.04 and 3.38 m. for RMSE,
as shown in Figure 5(b), Figure 5(c) and Figure 5(d) respectively.

Table 6 presents the statistical results of point comparisons between 10 DEM products compared with ICESat-2 ATL08,
indicating that the accuracy of JICA DEM and FABDEMv1-2 DEM was higher than other local and global DEM,
respectively. The statistical results of JICA DEM were -0.65 m, 1.04 m, 3.51, and 1.87 m for ME, MAE, MSE, and RMSE,
respectively. Specifically, the FABDEMv1-2 DEM showed the highest accuracy, with ME, MAE, MSE, and RMSE values
of 0.25 m, 0.80 m, 3.79, and 1.95 m, respectively.

Figure 6 presents the histogram distribution of ME for 10 DEM products relative to ICESat-2 ATL08 benchmark. The
histogram distribution illustrates that the entire curve of both local and global DEMs shifts towards negative and positive
biases, respectively. These shifts indicate that local DEMs, including LDD DEM, JICA DEM, and merged LDD-JICA DEM,
underestimate the elevation of the ICESat-2 ATL08 benchmark, with ME averages of -1.30 m, -0.65 m, and -0.68 m,
respectively.

Conversely, the shifts observed in the histogram distribution of global DEMs, including ASTERv3 DEM, SRTMv3 DEM,
Merit DEM, GLO30 DEM, FABDEMv1-2 DEM, TanDEM-X DEM, and TanDEM-EDEM DEM, indicate an overestimation
of the elevation of ICESat-2 ATL08 benchmark. The ME averages for these DEMs were +4.78 m, +2.03 m, +1.56 m, +0.84
m, +0.25 m, +0.94 m, and +0.91 m, respectively.



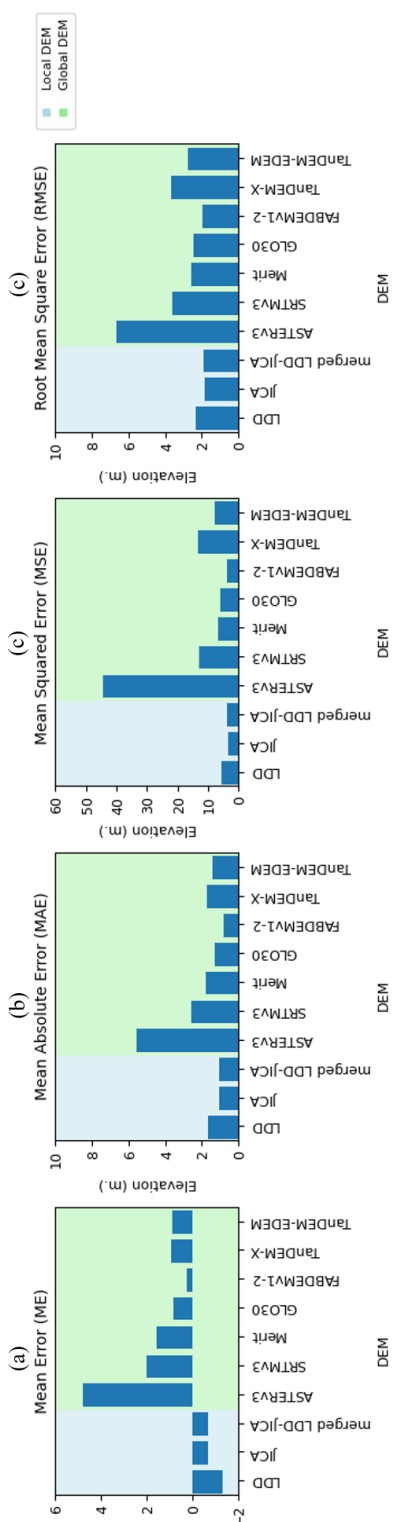

**Figure 5: a bar chart of statistical metrics, comparing 10 DEM products against the ICESat-2 ATL08 benchmark, including (a) mean error (ME), (b) mean absolute error (MAE), (c) mean squared error (MSE), and (d) root mean square error (RMSE). The resulting averages are computed across the datasets in study area.**

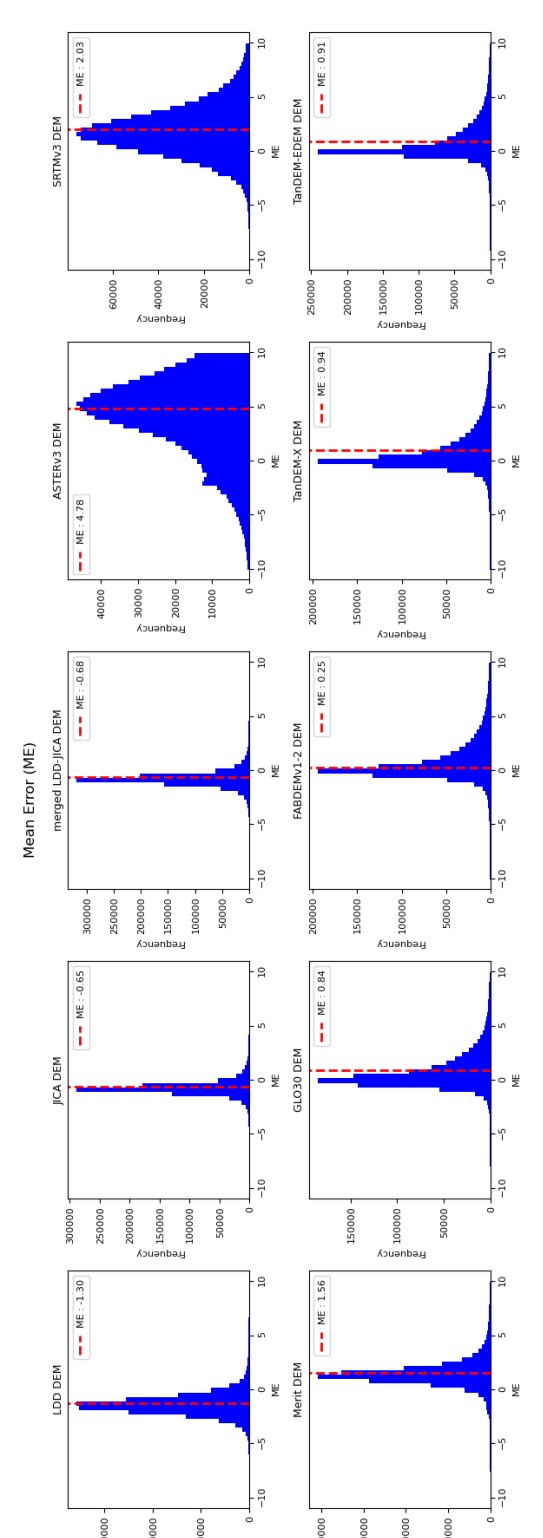

**Figure 6: histogram distribution of the mean error (ME), comparing 10 DEM products against the ICESat-2 ATL08 benchmark.**





**Table 6: Table of statistical metrics, comparing 10 DEM products against the ICESat-2 benchmark. The resulting averages are computed across the datasets in study area.**

| DEM product | Statistical method | | | |
|---|---|---|---|---|
| | ME (m.) | MAE (m.) | MSE (m.) | RMSE (m.) |
| LDD | -1.3 | 1.64 | 5.45 | 2.33 |
| JICA | -0.65 | 1.04 | 3.51 | 1.87 |
| merged LDD-JICA | -0.68 | 1.08 | 3.74 | 1.93 |
| ASTER | 4.77 | 5.57 | 44.28 | 6.65 |
| SRTM | 2.04 | 2.58 | 12.92 | 3.59 |
| MERIT | 1.56 | 1.79 | 6.76 | 2.6 |
| GlO30 | 0.84 | 1.3 | 5.89 | 2.43 |
| FABDEMv1-2 | 0.25 | 0.8 | 3.79 | 1.95 |
| TanDEM-X | 0.94 | 1.73 | 13.29 | 3.65 |
| TanDEM-EDEM | 0.91 | 1.43 | 7.74 | 2.78 |

### 5.2.2 Grid comparison evaluation results

Figure 7 displays the ME spatial grid comparison of 10 DEM products against the ICESat-2 ATL08 benchmark, with a resolution of 5x5 km. As shown in the figure, the local DEMs indicated overall lower values than the benchmark, with LDD DEM showing the lowest ME. In contrast, the overall ME spatial grid comparison of global DEMs was higher than the benchmark and clearly reveals that the most of global DEMs exhibit poor performance in urban areas. Notably, in the lower middle of the study area lies Bangkok, the capital city of Thailand. However, the FABDEMv1-2 DEM performed better in

urban areas compared to other global DEMs, which can be attribute to the improved algorithms described in Sect. 3.3.8 and Dandabathula et al., 2023.

### 5.2.3 Track-wise comparison evaluation results

The track-wise comparison involves comparing the land elevation profile over the study area between the 10 DEM products and ICESat-2 ATL08 benchmark (cf.Figure 8). As shown in Figure 8, it is evident that the local DEMs exhibit lower land

elevation compared to the ICESat-2 ATL08 benchmark. For most of the track, the LDD DEM measures a lower elevation than benchmark, while the JICA and merged LDD-JICA DEM follow the ICESat-2 ATL08 measurements more closely. This trend is consistent along the majority of the tracks, indicating that the LDD DEM generally underestimates the elevation compared to ICESat-2. Additionally, both the JICA and merged LDD-JICA DEMs closely track the ICESat-2 measurements for most of the tracks. Moreover, both of the local DEMs and ICESat-2 ATL08 can effectively remove buildings in urban

areas. However, there may be some points where the values cannot be entirely removed.

The overall result of the track-wise comparison of global DEMs shows a higher elevation than the benchmark, especially in urban areas, clearly indicating the shape of a hill in these urban areas, as illustrated in Figure 8. In the overall tracks, ASTERv3 and SRTMv3 DEMs tend to overestimate and exhibit fluctuations compared to the benchmark. Meanwhile, Merit, GLO30, TanDEM-X, and TanDEM-EDEM DEMs tend to follow a fluctuating pattern and measure slightly higher than the



**Figure 7: the mean error (ME) spatial grid comparison of 10 DEM products against the ICESat-2 ATL08 benchmark, with a resolution of 5x5 km.**



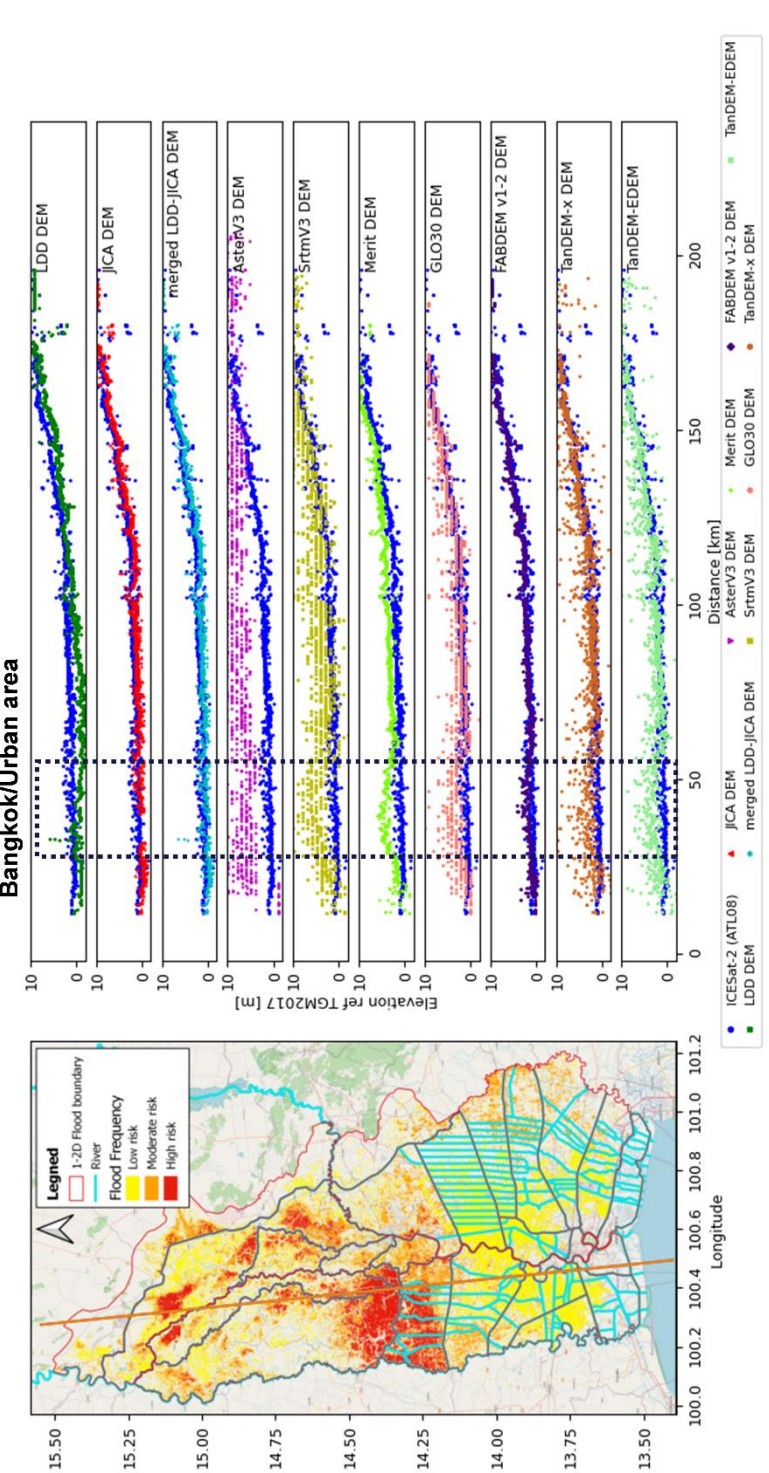

**Figure 8: The track-wise comparison of 10 DEM products with ICESat-2 ATL08 benchmark. © OpenStreetMap contributors 2015. Distributed under the Open Data Commons Open Database License (ODbL) v1.0.**





benchmark's track. FABDEMv1-2 closely aligns with the benchmark, indicating its strong performance. The track-wise comparison provides more detailed information in Appendix A.

The summary results of the evaluation of the 10 DEM products are presented in the parallel plot shown Figure 9, which displays the 10 DEM products along with the results of statistical methods including MAE, RMSE, and DEM resolution. In the local DEM products, it is notable that the LDD DEM exhibits higher error and resolution compared to the JICA and merged LDD-JICA DEMs. Both the JICA and merged LDD-JICA DEMs demonstrate similar accuracy, but the JICA DEM does not cover the entire study area (Figure 2). Therefore, we utilized the merged LDD-JICA DEM from the local DEM product to implement the 1D-2D flood model. For the global DEM product, the FABDEMv1-2 demonstrates the best performance compared to other global DEM products. Therefore, we selected the FABDEMv1-2 DEM to implement in the 1D-2D flood modelling, even though its spatial resolution is lower than TanDEM-X DEM.

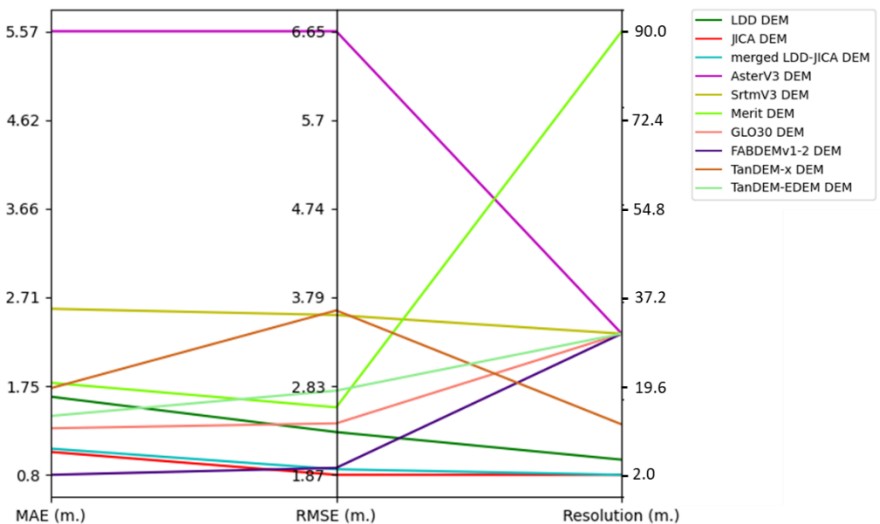

**Figure 9: The parallel plot of 10 DEMs evaluation with the ICESat-2 ATL08 benchmark.**

**5.3 Results of the evaluation of flood inundation maps**

We implemented the merged LDD-JICA DEM and the FABDEMv1-2 DEM from Sect. 5.2, into the 1D-2D flood model. The simulated flood map generated by the 1D-2D flood model, referred to as the Model flood map, was evaluated using flood maps from WorldWater and GISTDA for September, October, and November (flood season) in the years 2017 and 2021. The 1D-2D flood model generated daily simulated flood maps. To ensure accurate comparisons, we selected the dates of satellite passes over the study area according to Table A 1 and Table A 2. These dates were then combined to represent the flood areas that occurred in each month. The results of the flood map evaluation were categorized based on the DEM used and compared to the flood maps from WorldWater and GISTDA.

Table 7 provides a comparison of the POD, FAR, and CSI scores for the flood simulation using the merged LDD-JICA DEM, month, and year. Overall, the flood model using the merged LDD-JICA DEM tends to overestimate flooding,





particularly in the eastern part of the study area. This overestimation in the eastern part of the study area was attributed to the boundary between the JICA and LDD DEMs in the merged LDD-JICA DEM. The average FAR values of 0.926 and 0.790,

along with POD values of 0.713 and 0.585 compared to WorldWater and GISTDA flood maps, respectively, indicate that the Model flood map portrays a larger flood extent while still effectively detecting floods. The average CSI values of 0.072 and 0.183 indicate low model performance and a reflection of the larger flood extent simulation when compared to the flood maps by WorldWater and GISDTA

Figure 10 shows flood maps and contingency tables for September, October, and November in 2017 and 2021. Figure 10(a-

1) presents contingency tables comparing WorldWater monthly SWE and Model flood maps respectively. The results of the evaluation show low CSI values of 0.046, 0.071, and 0.076 for September, October, and November, respectively, indicating that the Model flood map has low performance. Additionally, the number of False alarms was high, resulting in high FAR values of 0.952, 0.926, and 0.923 for September, October, and November, respectively. Figure 10(b-1) illustrates contingency tables comparing GISTDA and Model flood maps in 2017. The POD values of 0.259, 0.567, and 0.642 are due

to the high number of Misses, particularly in September in the upper part of the study area. Moreover, the results show more false alarms in the eastern part of the study area, attributed to the combination of LDD and JICA DEMs. The FAR values of 0.913, 0.727, and 0.699 for September, October, and November, respectively. The CSI values were low in September at 0.070 but increased to 0.226 and 0.258 for October and November, respectively. The detailed statistics are summarized in Table 7.

Figure 10(a-2) and Figure 10(b-2) present contingency tables comparing WorldWater and Model, and GISTDA and Model for each month in 2021, respectively. The results of flood map evaluation in 2021 followed a similar trend to that of the 2017 flood. In Figure 10(a-2), low CSI values of 0.091, 0.071, and 0.075 are depicted for September, October, and December, respectively. Additionally, FAR values of 0.903, 0.928, and 0.923, and POD values of 0.593, 0.845, and 0.835, were high observed for September, October, and November, respectively. These values suggest that the WorldWater flood map

indicates a smaller flood extent compared to the Model flood map. Figure 10(b-2) illustrates an increase in CSI values to 0.133, 0.214, and 0.200 for September, October, and November, respectively, confirming that the Model flood map fit the GISTDA flood map as well. However, the FAR values were high at 0.852, 0.760, and 0.790 for September, October, and November, respectively, indicating that the Model flood map shows overestimated flood extents. Despite this, the POD values of 0.564, 0.667, and 0.810 suggest that the Model flood map can effectively detect GISTDA flood map extents,

particularly in October and November.

Table 7 presents a comparison of the POD, FAR, and CSI scores for the flood simulation using FABDEMv1-2 DEM, month, and year. The overall flood map evaluation based on the FABDEMv1-2 DEM indicates that the Model flood map tends to overestimate, with average FAR values of 0.916 and 0.730 compared to WorldWater and GISTDA flood maps, respectively. Meanwhile, the average CSI values of 0.081 and 0.230 indicate low performance.

Figure 11 shows flood maps and contingency tables in 2017 and 2021. Figure 11(a-1) illustrates contingency tables comparing WorldWater and Model flood maps for each month in 2017. The evaluation results clearly indicate that the





Model flood tends to overestimate the extent of flooding, as evidenced by FAR values of 0.946, 0.913, and 0.914 and low CSI values of 0.052, 0.084, and 0.085 in September, October, and November, respectively. However, the POD values were high, with values of 0.625, 0.710, and 0.907 in September, October, and November, respectively, indicating that the Model

flood map can effectively detect the WorldWater flood map as well, as shown in Table 7. Figure 11(b-1) presents contingency tables comparing GISTDA and Model floods for each month in 2017. The Figure 11(b-1) confirms the observations made in Figure 11(a-1), indicating that the Model flood map tends to overestimate the extent of flooding compared to the GISTDA flood map. However, the FAR values decrease slightly to 0.834, 0.612, and 0.591, and the POD values decrease to 0.331, 0.664, and 0.672 in September, October, and November, respectively. The decrease in POD is

attributed to a higher number of Misses in the upper part of the study area, suggesting that the GISTDA flood map depicts more flooding than the Model flood map. On the other hand, the CSI improved to 0.124, 0.325, and 0.341 in September, October, and November, respectively, indicating that the model results are more accurate when compared with GISTDA flood map. Additionally, the figure illustrates that the GISTDA flood map shows a greater extent of flooding compared to the WorldWater flood map.

Figure 11(a-2) and Figure 11(b-2) depict contingency tables comparing WorldWater and Model, and GISTDA and Model for each month in 2021, respectively. The Model flood map exhibits an overestimation of flooding, particularly noticeable in the eastern part of the study area. Figure 11(a-2) illustrates high FAR values of 0.887, 0.920, and 0.916 that indicating that there are more False alarms in September, October, and November, respectively. The POD was high values of 0.584, 0.885, and 0.850 and low CSI values of 0.105, 0.079, and 0.083 in September, October, and November, respectively. This figure

illustrates that the Model and the WorldWater flood map indicates more and less flooding, respectively. Figure 11(b-2) reveals more Misses in the upper part of the study area, resulting in a decrease in the POD values to 0.502, 0.680, and 0.837 compared to Figure 11(a-2). Despite this, the FAR values remain high at 0.832, 0.738, and 0.776, particularly notable in the eastern part of the study area. However, the Model flood map effectively detects the GISTDA flood map as well. The CSI values of 0.144, 0.234, and 0.215 for September, October, and November, respectively, indicate that the Model flood map

exhibits improved accuracy in comparison to the GISTDA flood map.

The overall assessment of the Model flood map, based on both the merged LDD-JICA and FABDEMv1-2 DEMs, indicates an overestimation of flood extent compared to both WorldWater and GISTDA flood maps. When comparing the model flood map based on the merged LDD-JICA DEM and FABDEMv1-2 DEM with each of the WorldWater and GISTDA flood maps, the results consistently indicate a slight improvement in performance for the Model flood map based on FABDEMv1-

2. The CSI of the Model flood map based on FABDEMv1-2 increases by 0.010 and 0.047 compared to the Model flood map based on the merged LDD-JICA DEM for WorldWater and GISTDA flood maps, respectively. Additionally, the FAR is reduced by approximately 0.010 and 0.060 for WorldWater and GISTDA flood maps, respectively. Although the study used flood classification processing to extract riverine flood maps from the SWE map for comparison, there are still limitations. Continuous improvement in flood classification process are necessary. The study results show that the overall assessment of

flood simulation based on FABDEMv1-2 DEM reveals a slight improvement of 13.55-25.56% in terms of the CSI compared



to flood simulation based on the merged LDD-JICA DEM. However, the DEM is one factor contributing to improved performance, many other factors still require further improvement.

**Table 7: The statistical metrics of the contingency table, comparing flood map dimensions 1 and 2.**

| Time | Dimension 1: WorldWater Dimension 2: Model | | | Dimension 1: GISTDA Dimension 2: Model | | | DEM product |
|---|---|---|---|---|---|---|---|
| | POD | FAR | CSI | POD | FAR | CSI | |
| 2017-09 | 0.549 | 0.952 | 0.046 | 0.259 | 0.913 | 0.070 | merged LDD-JICA |
| 2017-10 | 0.612 | 0.926 | 0.071 | 0.567 | 0.727 | 0.226 | merged LDD-JICA |
| 2017-11 | 0.842 | 0.923 | 0.076 | 0.642 | 0.699 | 0.258 | merged LDD-JICA |
| 2021-09 | 0.593 | 0.903 | 0.091 | 0.564 | 0.852 | 0.133 | merged LDD-JICA |
| 2021-10 | 0.845 | 0.928 | 0.071 | 0.667 | 0.760 | 0.214 | merged LDD-JICA |
| 2021-11 | 0.835 | 0.923 | 0.075 | 0.810 | 0.790 | 0.200 | merged LDD-JICA |
| **Total average** | **0.713** | **0.926** | **0.072** | **0.585** | **0.790** | **0.183** | **merged LDD-JICA** |
| 2017-09 | 0.625 | 0.946 | 0.052 | 0.331 | 0.834 | 0.124 | FABDEMv1-2 |
| 2017-10 | 0.710 | 0.913 | 0.084 | 0.664 | 0.612 | 0.325 | FABDEMv1-2 |
| 2017-11 | 0.907 | 0.914 | 0.085 | 0.672 | 0.591 | 0.341 | FABDEMv1-2 |
| 2021-09 | 0.584 | 0.887 | 0.105 | 0.502 | 0.832 | 0.144 | FABDEMv1-2 |
| 2021-10 | 0.885 | 0.920 | 0.079 | 0.680 | 0.738 | 0.234 | FABDEMv1-2 |
| 2021-11 | 0.850 | 0.916 | 0.083 | 0.837 | 0.776 | 0.215 | FABDEMv1-2 |
| **Total average** | **0.760** | **0.916** | **0.081** | **0.614** | **0.730** | **0.230** | **FABDEMv1-2** |

## 6 Discussion

### 6.1 Overall result of DEM analysis workflow

The result of DEM analysis shows that ICESat-2 ATL08 data offer a unique advantage in verifying DEM accuracy (Carabajal and Boy, 2020). The overall precision of DEM products was evaluated using the ICESat-2 ATL08 benchmark, showing that JICA and FAMDEMv1-2 DEM were significantly better than the local and global DEM products in terms average of RMSE, with values of 1.87 m and 1.95 m, respectively (Figure 5 and Table 6) in point comparison. The merged LDD-JICA DEM showed a slight difference of 0.06 m in average RMSE compared to the JICA DEM. This variance is primarily attributed to the combination of LDD and JICA DEMs, with JICA DEM being chosen as the primary DEM. However, it is noteworthy that the local DEM product exhibited a negative average bias (ME) ranging from -1.30 to -0.65 m, indicating that elevation of local DEM products is lower than the benchmark. Another study conducted in Spain, which verified Airborne LiDAR data with ICESat-2 ATL08, also reported a negative bias, with average ME values of -0.48 m (Zhu et al., 2022). On the other hand, the average ME of the global DEM products yielded positive values ranging from 0.25 to 4.77 m, indicating that the global DEM products overestimate the benchmark. This result has been previously confirmed in studies such as ASTERv3 (Weifeng et al., 2024), STRMv3 and TanDEM-X (Liu et al., 2020). The ASTERv3 DEM showed the lowest overall accuracy, with an average RMSE of 6.65 m. This is in line with other areas, such as the Qinghai-Tibet Plateau, where the RMSE reached 11.47 m (Weifeng et al., 2024). The TanDEM-EDEM is an updated version of TanDEM-X, which can reduce the error value from 3.65 to 2.78 m in terms of average RMSE.







**Figure 10: the contingency table of dimension 1 and dimension 2 flood maps on the spatial map: (a) Comparison between WorldWater and Model based on the merged LDD-JICA DEM, (a-1) in 2017 and (a-2) in 2021; (B) Comparison between GISTDA and Model based on the merged LDD-JICA DEM, (b-1) in 2017 and (b-2) in 2021.**



**Figure 11: the contingency table of dimension 1 and dimension 2 flood maps on the spatial map: (a) Comparison between WorldWater and Model based on FABDEMv1-2 DEM, (a-1) in 2017 and (a-2) in 2021; (b) Comparison between GISTDA and Model based on FABDEMv1-2 DEM, (b-1) in 2017 and (b-2) in 2021.**





In Figure 7, Figure A 11, and Figure A 12 illustrate the spatial grid comparison of 10 DEM products against the ICESat-2 ATL08 benchmark, with a resolution of 5x5 km for ME, MAE, and RMSE, respectively. The results clearly reveal that the global DEM tends to overestimate, particularly when compared to the ASTERv3 DEM. As shown in the figures, the error of
the global DEM clearly clusters in urban areas, except for the FABDEMv1-2, which employs an algorithm to remove building discrepancies, as discussed in Sect. 3.3.8. Although, ICESat-2 ATL08 is capable of measuring land elevation very accurately, some urban areas still exhibit overestimation, particularly in high-rise dense areas (Liu et al., 2020), as shown in Figure A 13. This suggests that the DEM analysis workflow can effectively utilize ICESat-2 ATL08 data for evaluation. In certain areas, the incorporation of ATL03 data may be necessary to enhance the evaluation process.

## 6.2 Overall result of flood map analysis workflow

The flood classification processing aims to classify flood types from SWE map. This method is based on various assumptions and simplifications. The validity of the approach is hard to evaluate, given the lack of ground-truth flood extent observations. However, it is evident that in this study area, surface water extent is not only due to riverine flooding but also various other flooding mechanisms such as irrigation and pluvial flooding.

The Model flood map, based on both Model and FABDEMv1-2 DEMs, tends to overestimate flood extent relative to the satellite EO datasets (Sect. 5.3). Additionally, the flood map based solely on FABDEMv1-2 performs slightly better than the one based on the merged LDD-JICA DEM, with an improvement of approximately 13.55 – 25.56 percent according to the CSI. The overestimation of flood inundation from the flood model occurs predominantly in the eastern part of the CPY River, indicating a clear need for improvement the 1D-2D flood model. Although this study has incorporated high-quality
DEM data implemented into the 1D-2D flood model, there are still many factors affecting flood map generation. For instance, the 1D-2D flood model, developed long ago (Sect. 3.1), needs to be updated and recalibrated due to continuous developments in water management plans, such as the Ayutthaya Bypass channel (JICA, 2018) and ongoing land use changes in the lower CPY basin (Visessri and Ekkawatpanit, 2020), which impact flood map simulations.

The results of the flood map comparisons demonstrate that the CSI value is relatively better when compared with GISTDA,
but lower when compared with WorldWater. It is observed that the overall WorldWater flood map shows relatively low flooding compared to the GISTDA flood map. This is due to fundamental differences in the mapping approaches with WorldWater aiming to provide long-time series of the typical distribution and persistence of monthly surface water presence whereas GISTDA is targeting real time maps showing the extent of flooding at a specific moment in time. Additionally, WorldWater uses only Sentinel-1 and Sentinel-2 data, whereas GISTDA combines data from multiple other satellites, as
described in Sect. 3.5.1. This can be further verified for accuracy with additional information from news sources and by cross-referencing with ICESat-2 ATL13 data, extracted from ICESat-2 ATL03 (inland water surface heights), in main rivers (Coppo Frias et al., 2023 and Dandabathula and Srinivasa Rao, 2020). This suggests that the flood analysis workflow can





effectively verify the performance of flood simulation using satellite data. Although this study's flood simulation results meet acceptable standards and are sufficiently reliable for practical applications.

## 7 Conclusion

The present study upgraded a 1D-2D flood model using satellite laser altimetry and multi-mission satellite SWE maps. We demonstrated two workflows in the lower CPY basin.

- **DEM analysis workflow:** This involved evaluating DEM accuracy using satellite laser altimetry data from ICESat-2 ATL08 before integrating the DEM products into the flood model. The assessment aimed to assess the overall performance of DEM products through vertical, spatial, track-wise analysis, and statistics measures to select the most suitable DEM for the study area. Furthermore, this workflow is transferable to other study areas, providing a method to reduce uncertainty before developing flood models. The results show that the merged LDD-JICA and FABDEMv1-2 DEMs are highly suitable in the study area, with RMSE values of 1.93 and 1.95 m., respectively.

- **Flood map analysis workflow:** This workflow encompassed riverine flood classification and the evaluation of simulated flood maps generated by the 1D-2D flood model using multi-mission satellite SWE maps. While the flood classification algorithm still presents challenges, it is important to recognize that SWE maps derived from satellite EO cannot be directly compared with the output of flood models without further processing. The flood map evaluation method facilitated the assessment of flood simulation accuracy against satellite SWE maps, employing statistical and spatial analyses. These evaluations contribute significantly to the calibration and validation of flood maps derived from the 1D-2D flood model. The results indicate that simulated flood maps based on FABDEMv1-2 DEM can improve the performance of the 1D-2D flood model by 13.55% to 25.56%, as determined by the CSI, when compared to simulated flood maps based on the merged LDD-JICA DEM.

Integrating these workflows will enhance the efficiency of the 1D-2D flood model and showcase the potential of utilizing EO satellite data to enhance flood modelling capabilities for operational flood forecasting in Thailand and elsewhere.





**Appendix A**

**Table A 1: Detail of GISTDA's flood map in study area**

| Year | Month | Day | Satellite |
|---|---|---|---|
| 2013 | November | 2 | |
| | | 3 | RADARSAT-2 |
| | | 7 | |
| | | 12 | |
| | | 16 | |
| 2014 | August | 31 | RADARSAT-2 |
| | September | 1 | RADARSAT-2 |
| | | 7 | COSMO-SkyMed1 |
| | | 8 | COSMO-SkyMed2, RADARSAT-2 |
| | | 10 | COSMO-SkyMed1, RADARSAT-2 |
| | | 11 | RADARSAT-2 |
| | | 12 | COSMO-SkyMed1 |
| | | 19 | COSMO-SkyMed1 |
| 2016 | October | 3 | COSMO-SkyMed4 |
| | | 5 | COSMO-SkyMed2 |
| | | 8 | RADARSAT-2 |
| | | 10 | COSMO-SkyMed4 |
| | | 13 | COSMO-SkyMed1 |
| | | 14 | RADARSAT-2 |
| | | 15 | RADARSAT-2 |
| | | 17 | RADARSAT-2 |
| | | 18 | RADARSAT-2 |
| | | 20 | COSMO-SkyMed1 |
| | | 21 | COSMO-SkyMed4 |
| | | 24 | RADARSAT-2 |
| | | 31 | COSMO-SkyMed2 |
| | November | 1 | RADARSAT-2 |
| | | 5 | COSMO-SkyMed4 |
| | | 11 | RADARSAT-2 |
| | | 21 | COSMO-SkyMed4 |
| | | 22 | Sentinel 8 |
| | December | 5 | RADARSAT-2 |
| 2017 | June | 11 | |
| | July | 23 | |
| | | 26 | COSMO-SkyMed2 |
| | | 28 | COSMO-SkyMed1 |
| | | 29 | RADARSAT-2 |
| | | 31 | COSMO-SkyMed4 |
| | August | 2 | |
| | | 5 | |
| | | 6 | |
| | | 8 | |
| | | 20 | |
| | | 22 | RADARSAT-2 |





| Year | Month | Day | Satellite |
|------|-------|-----|-----------|
|  |  | 25 | RADARSAT-2 |
|  |  | 26 | COSMO-SkyMed2 |
|  | September | 2 |  |
|  |  | 7 |  |
|  |  | 8 |  |
|  |  | 9 |  |
|  |  | 15 |  |
|  |  | 17 | RADARSAT-1 |
|  |  | 25 |  |
|  |  | 30 | RADARSAT-2 |
|  | October | 3 | Sentinel 8 |
|  |  | 8 |  |
|  |  | 12 |  |
|  |  | 13 | COSMO-SkyMed4 |
|  |  | 17 |  |
|  |  | 19 | COSMO<-SkyMed4 |
|  |  | 23 |  |
|  |  | 26 |  |
|  |  | 27 |  |
|  | November | 6 |  |
|  |  | 20 |  |
|  |  | 25 |  |
|  |  | 30 |  |
| 2021 | November | 2 |  |
|  |  | 3 |  |
|  |  | 8 |  |
|  |  | 9 |  |
|  |  | 14 |  |
|  |  | 15 |  |
|  | October | 3 |  |
|  |  | 4 |  |
|  |  | 6 |  |
|  |  | 8 |  |
|  |  | 12 |  |
|  |  | 14 |  |
|  |  | 16 |  |
|  |  | 17 |  |
|  |  | 18 |  |
|  |  | 21 |  |
|  |  | 22 |  |
|  |  | 27 |  |
|  |  | 28 |  |
|  | September | 12 |  |
|  |  | 21 |  |
|  |  | 22 |  |
|  |  | 27 |  |
|  |  | 28 |  |





| Year | Month | Day | Satellite |
|---|---|---|---|
| | | 29 | |

**Table A 2: The detailed analysis of Sentinel-1 and Sentinel-2 data within the study area.**

| Year | Month | Day | Satellite | Year | Month | Day | Satellite |
|---|---|---|---|---|---|---|---|
| 2017 | 1 | 2 | S1 | 2017 | 5 | 7 | S2A |
| 2017 | 1 | 14 | S1 | 2017 | 5 | 10 | S1 |
| 2017 | 1 | 16 | S1 | 2017 | 5 | 10 | S2A |
| 2017 | 1 | 20 | S2A | 2017 | 5 | 14 | S1 |
| 2017 | 1 | 27 | S2A | 2017 | 5 | 17 | S2A |
| 2017 | 2 | 7 | S1 | 2017 | 5 | 22 | S1 |
| 2017 | 2 | 9 | S2A | 2017 | 5 | 26 | S1 |
| 2017 | 2 | 16 | S2A | 2017 | 5 | 27 | S2A |
| 2017 | 2 | 19 | S2A | 2017 | 5 | 30 | S2A |
| 2017 | 2 | 26 | S2A | 2017 | 6 | 3 | S1 |
| 2017 | 2 | 27 | S1 | 2017 | 6 | 6 | S2A |
| 2017 | 3 | 8 | S2A | 2017 | 6 | 7 | S1 |
| 2017 | 3 | 11 | S1 | 2017 | 6 | 15 | S1 |
| 2017 | 3 | 11 | S2A | 2017 | 6 | 16 | S2A |
| 2017 | 3 | 15 | S1 | 2017 | 6 | 19 | S1 |
| 2017 | 3 | 18 | S2A | 2017 | 6 | 19 | S2A |
| 2017 | 3 | 23 | S1 | 2017 | 6 | 26 | S2A |
| 2017 | 3 | 27 | S1 | 2017 | 6 | 27 | S1 |
| 2017 | 3 | 28 | S2A | 2017 | 7 | 1 | S1 |
| 2017 | 3 | 31 | S2A | 2017 | 7 | 6 | S2A |
| 2017 | 4 | 4 | S1 | 2017 | 7 | 9 | S1 |
| 2017 | 4 | 7 | S2A | 2017 | 7 | 9 | S2A |
| 2017 | 4 | 8 | S1 | 2017 | 7 | 11 | S2B |
| 2017 | 4 | 16 | S1 | 2017 | 7 | 13 | S1 |
| 2017 | 4 | 17 | S2A | 2017 | 7 | 16 | S2A |
| 2017 | 4 | 20 | S1 | 2017 | 7 | 21 | S1 |
| 2017 | 4 | 20 | S2A | 2017 | 7 | 21 | S2B |
| 2017 | 4 | 27 | S2A | 2017 | 7 | 24 | S2B |
| 2017 | 4 | 28 | S1 | 2017 | 7 | 25 | S1 |
| 2017 | 5 | 2 | S1 | 2017 | 7 | 26 | S2A |
| 2017 | 7 | 29 | S2A | 2017 | 10 | 25 | S1 |
| 2017 | 7 | 31 | S2B | 2017 | 11 | 1 | S2B |
| 2017 | 8 | 2 | S1 | 2017 | 11 | 3 | S2A |
| 2017 | 8 | 5 | S2A | 2017 | 11 | 6 | S1 |
| 2017 | 8 | 6 | S1 | 2017 | 11 | 6 | S2A |
| 2017 | 8 | 10 | S2B | 2017 | 11 | 8 | S2B |
| 2017 | 8 | 13 | S2B | 2017 | 11 | 10 | S1 |



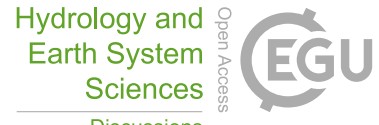

| Year | Month | Day | Satellite | Year | Month | Day | Satellite |
|------|-------|-----|-----------|------|-------|-----|-----------|
| 2017 | 8 | 14 | S1 | 2017 | 11 | 11 | S2B |
| 2017 | 8 | 15 | S2A | 2017 | 11 | 13 | S2A |
| 2017 | 8 | 18 | S1 | 2017 | 11 | 18 | S1 |
| 2017 | 8 | 18 | S2A | 2017 | 11 | 18 | S2B |
| 2017 | 8 | 20 | S2B | 2017 | 11 | 21 | S2B |
| 2017 | 8 | 25 | S2A | 2017 | 11 | 22 | S1 |
| 2017 | 8 | 26 | S1 | 2017 | 11 | 23 | S2A |
| 2017 | 8 | 30 | S1 | 2017 | 11 | 26 | S2A |
| 2017 | 8 | 30 | S2B | 2017 | 11 | 28 | S2B |
| 2017 | 9 | 2 | S2B | 2017 | 11 | 30 | S1 |
| 2017 | 9 | 4 | S2A | 2017 | 12 | 1 | S2B |
| 2017 | 9 | 7 | S1 | 2017 | 12 | 3 | S2A |
| 2017 | 9 | 7 | S2A | 2017 | 12 | 4 | S1 |
| 2017 | 9 | 9 | S2B | 2017 | 12 | 8 | S2B |
| 2017 | 9 | 11 | S1 | 2017 | 12 | 11 | S2B |
| 2017 | 9 | 14 | S2A | 2017 | 12 | 12 | S1 |
| 2017 | 9 | 19 | S1 | 2017 | 12 | 13 | S2A |
| 2017 | 9 | 19 | S2B | 2017 | 12 | 16 | S1 |
| 2017 | 9 | 22 | S2B | 2017 | 12 | 16 | S2A |
| 2017 | 9 | 23 | S1 | 2017 | 12 | 18 | S2B |
| 2017 | 9 | 24 | S2A | 2017 | 12 | 21 | S2B |
| 2017 | 9 | 27 | S2A | 2017 | 12 | 23 | S2A |
| 2017 | 9 | 29 | S2B | 2017 | 12 | 24 | S1 |
| 2017 | 10 | 4 | S2A | 2017 | 12 | 28 | S1 |
| 2017 | 10 | 5 | S1 | 2017 | 12 | 28 | S2B |
| 2017 | 10 | 9 | S2B | 2017 | 12 | 31 | S2B |
| 2017 | 10 | 12 | S2B | 2018 | 1 | 2 | S1 |
| 2017 | 10 | 13 | S1 | 2018 | 1 | 2 | S2A |
| 2017 | 10 | 14 | S2A | 2018 | 1 | 5 | S2A |
| 2017 | 10 | 17 | S1 | 2018 | 1 | 7 | S2B |
| 2017 | 10 | 17 | S2A | 2018 | 1 | 9 | S1 |
| 2017 | 10 | 19 | S2B | 2018 | 1 | 10 | S2B |
| 2017 | 10 | 24 | S2A | 2018 | 1 | 12 | S2A |
| 2018 | 1 | 17 | S1 | 2018 | 3 | 30 | S1 |
| 2018 | 1 | 17 | S2B | 2018 | 3 | 31 | S2B |
| 2018 | 1 | 20 | S2B | 2018 | 4 | 2 | S2A |
| 2018 | 1 | 21 | S1 | 2018 | 4 | 3 | S1 |
| 2018 | 1 | 22 | S2A | 2018 | 4 | 5 | S2A |
| 2018 | 1 | 25 | S2A | 2018 | 4 | 7 | S2B |
| 2018 | 1 | 27 | S2B | 2018 | 4 | 10 | S2B |
| 2018 | 1 | 29 | S1 | 2018 | 4 | 11 | S1 |





| Year | Month | Day | Satellite | Year | Month | Day | Satellite |
|------|-------|-----|-----------|------|-------|-----|-----------|
| 2018 | 1 | 30 | S2B | 2018 | 4 | 12 | S2A |
| 2018 | 2 | 1 | S2A | 2018 | 4 | 15 | S1 |
| 2018 | 2 | 2 | S1 | 2018 | 4 | 15 | S2A |
| 2018 | 2 | 6 | S2B | 2018 | 4 | 17 | S2B |
| 2018 | 2 | 9 | S2B | 2018 | 4 | 20 | S2B |
| 2018 | 2 | 10 | S1 | 2018 | 4 | 22 | S2A |
| 2018 | 2 | 11 | S2A | 2018 | 4 | 23 | S1 |
| 2018 | 2 | 14 | S1 | 2018 | 4 | 25 | S2A |
| 2018 | 2 | 14 | S2A | 2018 | 4 | 27 | S1 |
| 2018 | 2 | 16 | S2B | 2018 | 4 | 27 | S2B |
| 2018 | 2 | 19 | S2B | 2018 | 4 | 30 | S2B |
| 2018 | 2 | 21 | S2A | 2018 | 5 | 2 | S2A |
| 2018 | 2 | 22 | S1 | 2018 | 5 | 5 | S1 |
| 2018 | 2 | 24 | S2A | 2018 | 5 | 5 | S2A |
| 2018 | 2 | 26 | S1 | 2018 | 5 | 7 | S2B |
| 2018 | 2 | 26 | S2B | 2018 | 5 | 9 | S1 |
| 2018 | 3 | 1 | S2B | 2018 | 5 | 10 | S2B |
| 2018 | 3 | 3 | S2A | 2018 | 5 | 12 | S2A |
| 2018 | 3 | 6 | S1 | 2018 | 5 | 15 | S2A |
| 2018 | 3 | 6 | S2A | 2018 | 5 | 17 | S1 |
| 2018 | 3 | 8 | S2B | 2018 | 5 | 17 | S2B |
| 2018 | 3 | 10 | S1 | 2018 | 5 | 20 | S2B |
| 2018 | 3 | 11 | S2B | 2018 | 5 | 21 | S1 |
| 2018 | 3 | 13 | S2A | 2018 | 5 | 22 | S2A |
| 2018 | 3 | 16 | S2A | 2018 | 5 | 25 | S2A |
| 2018 | 3 | 18 | S1 | 2018 | 5 | 29 | S1 |
| 2018 | 3 | 18 | S2B | 2018 | 5 | 30 | S2B |
| 2018 | 3 | 21 | S2B | 2018 | 6 | 1 | S2A |
| 2018 | 3 | 22 | S1 | 2018 | 6 | 2 | S1 |
| 2018 | 3 | 23 | S2A | 2018 | 6 | 4 | S2A |
| 2018 | 3 | 26 | S2A | 2018 | 6 | 6 | S2B |
| 2018 | 3 | 28 | S2B | 2018 | 6 | 9 | S2B |
| 2018 | 6 | 10 | S1 | 2018 | 8 | 20 | S2A |
| 2018 | 6 | 11 | S2A | 2018 | 8 | 21 | S1 |
| 2018 | 6 | 14 | S1 | 2018 | 8 | 23 | S2A |
| 2018 | 6 | 14 | S2A | 2018 | 8 | 25 | S2B |
| 2018 | 6 | 16 | S2B | 2018 | 8 | 28 | S2B |
| 2018 | 6 | 19 | S2B | 2018 | 8 | 30 | S2A |
| 2018 | 6 | 21 | S2A | 2018 | 9 | 2 | S1 |
| 2018 | 6 | 22 | S1 | 2018 | 9 | 2 | S2A |
| 2018 | 6 | 24 | S2A | 2018 | 9 | 4 | S2B |





| Year | Month | Day | Satellite | Year | Month | Day | Satellite |
|---|---|---|---|---|---|---|---|
| 2018 | 6 | 26 | S1 | 2018 | 9 | 6 | S1 |
| 2018 | 6 | 26 | S2B | 2018 | 9 | 7 | S2B |
| 2018 | 6 | 29 | S2B | 2018 | 9 | 9 | S2A |
| 2018 | 7 | 1 | S2A | 2018 | 9 | 12 | S2A |
| 2018 | 7 | 4 | S1 | 2018 | 9 | 14 | S1 |
| 2018 | 7 | 4 | S2A | 2018 | 9 | 14 | S2B |
| 2018 | 7 | 6 | S2B | 2018 | 9 | 17 | S2B |
| 2018 | 7 | 8 | S1 | 2018 | 9 | 18 | S1 |
| 2018 | 7 | 9 | S2B | 2018 | 9 | 19 | S2A |
| 2018 | 7 | 11 | S2A | 2018 | 9 | 24 | S2B |
| 2018 | 7 | 14 | S2A | 2018 | 9 | 26 | S1 |
| 2018 | 7 | 16 | S1 | 2018 | 9 | 27 | S2B |
| 2018 | 7 | 16 | S2B | 2018 | 9 | 29 | S2A |
| 2018 | 7 | 19 | S2B | 2018 | 9 | 30 | S1 |
| 2018 | 7 | 20 | S1 | 2018 | 10 | 2 | S2A |
| 2018 | 7 | 21 | S2A | 2018 | 10 | 4 | S2B |
| 2018 | 7 | 24 | S2A | 2018 | 10 | 7 | S2B |
| 2018 | 7 | 26 | S2B | 2018 | 10 | 8 | S1 |
| 2018 | 7 | 28 | S1 | 2018 | 10 | 9 | S2A |
| 2018 | 7 | 29 | S2B | 2018 | 10 | 12 | S1 |
| 2018 | 7 | 31 | S2A | 2018 | 10 | 12 | S2A |
| 2018 | 8 | 1 | S1 | 2018 | 10 | 14 | S2B |
| 2018 | 8 | 3 | S2A | 2018 | 10 | 17 | S2B |
| 2018 | 8 | 5 | S2B | 2018 | 10 | 19 | S2A |
| 2018 | 8 | 8 | S2B | 2018 | 10 | 20 | S1 |
| 2018 | 8 | 9 | S1 | 2018 | 10 | 22 | S2A |
| 2018 | 8 | 10 | S2A | 2018 | 10 | 24 | S1 |
| 2018 | 8 | 13 | S1 | 2018 | 10 | 24 | S2B |
| 2018 | 8 | 13 | S2A | 2018 | 10 | 27 | S2B |
| 2018 | 8 | 15 | S2B | 2018 | 10 | 29 | S2A |
| 2018 | 8 | 18 | S2B | 2018 | 11 | 1 | S1 |
| 2018 | 11 | 1 | S2A | 2019 | 1 | 15 | S2B |
| 2018 | 11 | 3 | S2B | 2019 | 1 | 17 | S2A |
| 2018 | 11 | 5 | S1 | 2019 | 1 | 20 | S2A |
| 2018 | 11 | 6 | S2B | 2019 | 1 | 22 | S2B |
| 2018 | 11 | 8 | S2A | 2019 | 1 | 24 | S1 |
| 2018 | 11 | 11 | S2A | 2019 | 1 | 25 | S2B |
| 2018 | 11 | 13 | S1 | 2019 | 1 | 27 | S2A |
| 2018 | 11 | 16 | S2B | 2019 | 1 | 28 | S1 |
| 2018 | 11 | 17 | S1 | 2019 | 1 | 30 | S2A |
| 2018 | 11 | 18 | S2A | 2019 | 2 | 1 | S2B |





| Year | Month | Day | Satellite | Year | Month | Day | Satellite |
|------|-------|-----|-----------|------|-------|-----|-----------|
| 2018 | 11 | 21 | S2A | 2019 | 2 | 4 | S2B |
| 2018 | 11 | 23 | S2B | 2019 | 2 | 5 | S1 |
| 2018 | 11 | 25 | S1 | 2019 | 2 | 6 | S2A |
| 2018 | 11 | 26 | S2B | 2019 | 2 | 9 | S1 |
| 2018 | 11 | 28 | S2A | 2019 | 2 | 9 | S2A |
| 2018 | 11 | 29 | S1 | 2019 | 2 | 11 | S2B |
| 2018 | 12 | 1 | S2A | 2019 | 2 | 14 | S2B |
| 2018 | 12 | 3 | S2B | 2019 | 2 | 16 | S2A |
| 2018 | 12 | 6 | S2B | 2019 | 2 | 17 | S1 |
| 2018 | 12 | 7 | S1 | 2019 | 2 | 19 | S2A |
| 2018 | 12 | 8 | S2A | 2019 | 2 | 21 | S1 |
| 2018 | 12 | 11 | S1 | 2019 | 2 | 21 | S2B |
| 2018 | 12 | 11 | S2A | 2019 | 2 | 24 | S2B |
| 2018 | 12 | 13 | S2B | 2019 | 2 | 26 | S2A |
| 2018 | 12 | 16 | S2B | 2019 | 3 | 1 | S1 |
| 2018 | 12 | 18 | S2A | 2019 | 3 | 1 | S2A |
| 2018 | 12 | 19 | S1 | 2019 | 3 | 3 | S2B |
| 2018 | 12 | 21 | S2A | 2019 | 3 | 5 | S1 |
| 2018 | 12 | 23 | S1 | 2019 | 3 | 6 | S2B |
| 2018 | 12 | 23 | S2B | 2019 | 3 | 8 | S2A |
| 2018 | 12 | 26 | S2B | 2019 | 3 | 11 | S2A |
| 2018 | 12 | 28 | S2A | 2019 | 3 | 13 | S1 |
| 2018 | 12 | 31 | S1 | 2019 | 3 | 13 | S2B |
| 2018 | 12 | 31 | S2A | 2019 | 3 | 16 | S2B |
| 2019 | 1 | 2 | S2B | 2019 | 3 | 18 | S2A |
| 2019 | 1 | 4 | S1 | 2019 | 3 | 21 | S2A |
| 2019 | 1 | 5 | S2B | 2019 | 3 | 23 | S2B |
| 2019 | 1 | 7 | S2A | 2019 | 3 | 25 | S1 |
| 2019 | 1 | 10 | S2A | 2019 | 3 | 26 | S2B |
| 2019 | 1 | 12 | S1 | 2019 | 3 | 28 | S2A |
| 2019 | 3 | 29 | S1 | 2019 | 6 | 11 | S2B |
| 2019 | 3 | 31 | S2A | 2019 | 6 | 14 | S2B |
| 2019 | 4 | 2 | S2B | 2019 | 6 | 16 | S2A |
| 2019 | 4 | 5 | S2B | 2019 | 6 | 17 | S1 |
| 2019 | 4 | 6 | S1 | 2019 | 6 | 19 | S2A |
| 2019 | 4 | 7 | S2A | 2019 | 6 | 21 | S2B |
| 2019 | 4 | 10 | S1 | 2019 | 6 | 24 | S2B |
| 2019 | 4 | 10 | S2A | 2019 | 6 | 26 | S2A |
| 2019 | 4 | 12 | S2B | 2019 | 6 | 29 | S1 |
| 2019 | 4 | 15 | S2B | 2019 | 6 | 29 | S2A |
| 2019 | 4 | 17 | S2A | 2019 | 7 | 1 | S2B |





| Year | Month | Day | Satellite | Year | Month | Day | Satellite |
|------|-------|-----|-----------|------|-------|-----|-----------|
| 2019 | 4 | 20 | S2A | 2019 | 7 | 4 | S2B |
| 2019 | 4 | 22 | S1 | 2019 | 7 | 6 | S2A |
| 2019 | 4 | 22 | S2B | 2019 | 7 | 9 | S2A |
| 2019 | 4 | 25 | S2B | 2019 | 7 | 11 | S1 |
| 2019 | 4 | 27 | S2A | 2019 | 7 | 11 | S2B |
| 2019 | 4 | 30 | S1 | 2019 | 7 | 14 | S2B |
| 2019 | 4 | 30 | S2A | 2019 | 7 | 16 | S2A |
| 2019 | 5 | 2 | S2B | 2019 | 7 | 19 | S2A |
| 2019 | 5 | 4 | S1 | 2019 | 7 | 21 | S2B |
| 2019 | 5 | 5 | S2B | 2019 | 7 | 23 | S1 |
| 2019 | 5 | 7 | S2A | 2019 | 7 | 24 | S2B |
| 2019 | 5 | 10 | S2A | 2019 | 7 | 26 | S2A |
| 2019 | 5 | 12 | S1 | 2019 | 7 | 29 | S2A |
| 2019 | 5 | 12 | S2B | 2019 | 7 | 31 | S2B |
| 2019 | 5 | 15 | S2B | 2019 | 8 | 3 | S2B |
| 2019 | 5 | 16 | S1 | 2019 | 8 | 4 | S1 |
| 2019 | 5 | 17 | S2A | 2019 | 8 | 5 | S2A |
| 2019 | 5 | 20 | S2A | 2019 | 8 | 8 | S2A |
| 2019 | 5 | 22 | S2B | 2019 | 8 | 10 | S2B |
| 2019 | 5 | 24 | S1 | 2019 | 8 | 13 | S2B |
| 2019 | 5 | 25 | S2B | 2019 | 8 | 15 | S2A |
| 2019 | 5 | 27 | S2A | 2019 | 8 | 16 | S1 |
| 2019 | 5 | 28 | S1 | 2019 | 8 | 18 | S2A |
| 2019 | 5 | 30 | S2A | 2019 | 8 | 20 | S1 |
| 2019 | 6 | 1 | S2B | 2019 | 8 | 20 | S2B |
| 2019 | 6 | 4 | S2B | 2019 | 8 | 23 | S2B |
| 2019 | 6 | 5 | S1 | 2019 | 8 | 25 | S2A |
| 2019 | 6 | 6 | S2A | 2019 | 8 | 28 | S1 |
| 2019 | 6 | 9 | S2A | 2019 | 8 | 28 | S2A |
| 2019 | 8 | 30 | S2B | 2019 | 11 | 11 | S2B |
| 2019 | 9 | 1 | S1 | 2019 | 11 | 12 | S1 |
| 2019 | 9 | 2 | S2B | 2019 | 11 | 13 | S2A |
| 2019 | 9 | 4 | S2A | 2019 | 11 | 16 | S2A |
| 2019 | 9 | 7 | S2A | 2019 | 11 | 18 | S2B |
| 2019 | 9 | 9 | S1 | 2019 | 11 | 20 | S1 |
| 2019 | 9 | 9 | S2B | 2019 | 11 | 21 | S2B |
| 2019 | 9 | 12 | S2B | 2019 | 11 | 23 | S2A |
| 2019 | 9 | 13 | S1 | 2019 | 11 | 24 | S1 |
| 2019 | 9 | 14 | S2A | 2019 | 11 | 26 | S2A |
| 2019 | 9 | 17 | S2A | 2019 | 11 | 28 | S2B |
| 2019 | 9 | 19 | S2B | 2019 | 12 | 1 | S2B |





| Year | Month | Day | Satellite | Year | Month | Day | Satellite |
|------|-------|-----|-----------|------|-------|-----|-----------|
| 2019 | 9 | 21 | S1 | 2019 | 12 | 2 | S1 |
| 2019 | 9 | 22 | S2B | 2019 | 12 | 3 | S2A |
| 2019 | 9 | 25 | S1 | 2019 | 12 | 6 | S1 |
| 2019 | 9 | 27 | S2A | 2019 | 12 | 6 | S2A |
| 2019 | 9 | 29 | S2B | 2019 | 12 | 8 | S2B |
| 2019 | 10 | 2 | S2B | 2019 | 12 | 11 | S2B |
| 2019 | 10 | 3 | S1 | 2019 | 12 | 13 | S2A |
| 2019 | 10 | 4 | S2A | 2019 | 12 | 14 | S1 |
| 2019 | 10 | 7 | S1 | 2019 | 12 | 16 | S2A |
| 2019 | 10 | 7 | S2A | 2019 | 12 | 18 | S2B |
| 2019 | 10 | 9 | S2B | 2019 | 12 | 21 | S2B |
| 2019 | 10 | 12 | S2B | 2019 | 12 | 23 | S2A |
| 2019 | 10 | 14 | S2A | 2019 | 12 | 26 | S1 |
| 2019 | 10 | 15 | S1 | 2019 | 12 | 26 | S2A |
| 2019 | 10 | 17 | S2A | 2019 | 12 | 28 | S2B |
| 2019 | 10 | 19 | S1 | 2019 | 12 | 30 | S1 |
| 2019 | 10 | 19 | S2B | 2019 | 12 | 31 | S2B |
| 2019 | 10 | 22 | S2B | 2020 | 1 | 2 | S2A |
| 2019 | 10 | 24 | S2A | 2020 | 1 | 5 | S2A |
| 2019 | 10 | 27 | S1 | 2020 | 1 | 7 | S1 |
| 2019 | 10 | 27 | S2A | 2020 | 1 | 7 | S2B |
| 2019 | 10 | 29 | S2B | 2020 | 1 | 10 | S2B |
| 2019 | 10 | 31 | S1 | 2020 | 1 | 11 | S1 |
| 2019 | 11 | 1 | S2B | 2020 | 1 | 12 | S2A |
| 2019 | 11 | 3 | S2A | 2020 | 1 | 15 | S2A |
| 2019 | 11 | 6 | S2A | 2020 | 1 | 17 | S2B |
| 2019 | 11 | 8 | S1 | 2020 | 1 | 18 | S1 |
| 2019 | 11 | 8 | S2B | 2020 | 1 | 19 | S1 |
| 2020 | 1 | 20 | S2B | 2020 | 3 | 31 | S1 |
| 2020 | 1 | 22 | S2A | 2020 | 4 | 1 | S2A |
| 2020 | 1 | 23 | S1 | 2020 | 4 | 4 | S1 |
| 2020 | 1 | 25 | S2A | 2020 | 4 | 4 | S2A |
| 2020 | 1 | 27 | S2B | 2020 | 4 | 6 | S2B |
| 2020 | 1 | 30 | S2B | 2020 | 4 | 9 | S2B |
| 2020 | 1 | 31 | S1 | 2020 | 4 | 11 | S2A |
| 2020 | 2 | 1 | S2A | 2020 | 4 | 12 | S1 |
| 2020 | 2 | 4 | S1 | 2020 | 4 | 14 | S2A |
| 2020 | 2 | 4 | S2A | 2020 | 4 | 16 | S1 |
| 2020 | 2 | 6 | S2B | 2020 | 4 | 16 | S2B |
| 2020 | 2 | 9 | S2B | 2020 | 4 | 19 | S2B |
| 2020 | 2 | 11 | S2A | 2020 | 4 | 21 | S2A |



| Year | Month | Day | Satellite | Year | Month | Day | Satellite |
|------|-------|-----|-----------|------|-------|-----|-----------|
| 2020 | 2 | 12 | S1 | 2020 | 4 | 24 | S1 |
| 2020 | 2 | 14 | S2A | 2020 | 4 | 24 | S2A |
| 2020 | 2 | 16 | S1 | 2020 | 4 | 26 | S2B |
| 2020 | 2 | 16 | S2B | 2020 | 4 | 28 | S1 |
| 2020 | 2 | 19 | S2B | 2020 | 4 | 29 | S2B |
| 2020 | 2 | 21 | S2A | 2020 | 5 | 1 | S2A |
| 2020 | 2 | 24 | S1 | 2020 | 5 | 4 | S2A |
| 2020 | 2 | 24 | S2A | 2020 | 5 | 6 | S1 |
| 2020 | 2 | 26 | S2B | 2020 | 5 | 6 | S2B |
| 2020 | 2 | 28 | S1 | 2020 | 5 | 9 | S2B |
| 2020 | 2 | 29 | S2B | 2020 | 5 | 11 | S2A |
| 2020 | 3 | 2 | S2A | 2020 | 5 | 14 | S2A |
| 2020 | 3 | 5 | S2A | 2020 | 5 | 16 | S2B |
| 2020 | 3 | 7 | S1 | 2020 | 5 | 18 | S1 |
| 2020 | 3 | 7 | S2B | 2020 | 5 | 19 | S2B |
| 2020 | 3 | 10 | S2B | 2020 | 5 | 21 | S2A |
| 2020 | 3 | 11 | S1 | 2020 | 5 | 24 | S2A |
| 2020 | 3 | 12 | S2A | 2020 | 5 | 26 | S2B |
| 2020 | 3 | 15 | S2A | 2020 | 5 | 29 | S2B |
| 2020 | 3 | 17 | S2B | 2020 | 5 | 30 | S1 |
| 2020 | 3 | 19 | S1 | 2020 | 5 | 31 | S2A |
| 2020 | 3 | 20 | S2B | 2020 | 6 | 3 | S2A |
| 2020 | 3 | 22 | S2A | 2020 | 6 | 5 | S2B |
| 2020 | 3 | 23 | S1 | 2020 | 6 | 8 | S2B |
| 2020 | 3 | 25 | S2A | 2020 | 6 | 10 | S2A |
| 2020 | 3 | 27 | S2B | 2020 | 6 | 11 | S1 |
| 2020 | 3 | 30 | S2B | 2020 | 6 | 13 | S2A |
| 2020 | 6 | 15 | S2B | 2020 | 9 | 3 | S1 |
| 2020 | 6 | 18 | S2B | 2020 | 9 | 3 | S2B |
| 2020 | 6 | 20 | S2A | 2020 | 9 | 6 | S2B |
| 2020 | 6 | 23 | S1 | 2020 | 9 | 7 | S1 |
| 2020 | 6 | 23 | S2A | 2020 | 9 | 8 | S2A |
| 2020 | 6 | 25 | S2B | 2020 | 9 | 11 | S2A |
| 2020 | 6 | 28 | S2B | 2020 | 9 | 13 | S2B |
| 2020 | 6 | 30 | S2A | 2020 | 9 | 15 | S1 |
| 2020 | 7 | 3 | S2A | 2020 | 9 | 16 | S2B |
| 2020 | 7 | 5 | S1 | 2020 | 9 | 18 | S2A |
| 2020 | 7 | 5 | S2B | 2020 | 9 | 19 | S1 |
| 2020 | 7 | 8 | S2B | 2020 | 9 | 21 | S2A |
| 2020 | 7 | 10 | S2A | 2020 | 9 | 23 | S2B |
| 2020 | 7 | 13 | S2A | 2020 | 9 | 26 | S2B |





| Year | Month | Day | Satellite | Year | Month | Day | Satellite |
|---|---|---|---|---|---|---|---|
| 2020 | 7 | 15 | S2B | 2020 | 9 | 27 | S1 |
| 2020 | 7 | 17 | S1 | 2020 | 9 | 28 | S2A |
| 2020 | 7 | 18 | S2B | 2020 | 10 | 1 | S1 |
| 2020 | 7 | 20 | S2A | 2020 | 10 | 1 | S2A |
| 2020 | 7 | 23 | S2A | 2020 | 10 | 3 | S2B |
| 2020 | 7 | 25 | S2B | 2020 | 10 | 6 | S2B |
| 2020 | 7 | 28 | S2B | 2020 | 10 | 8 | S2A |
| 2020 | 7 | 29 | S1 | 2020 | 10 | 9 | S1 |
| 2020 | 7 | 30 | S2A | 2020 | 10 | 11 | S2A |
| 2020 | 8 | 2 | S2A | 2020 | 10 | 13 | S1 |
| 2020 | 8 | 4 | S2B | 2020 | 10 | 13 | S2B |
| 2020 | 8 | 7 | S2B | 2020 | 10 | 16 | S2B |
| 2020 | 8 | 9 | S2A | 2020 | 10 | 18 | S2A |
| 2020 | 8 | 10 | S1 | 2020 | 10 | 21 | S1 |
| 2020 | 8 | 12 | S2A | 2020 | 10 | 21 | S2A |
| 2020 | 8 | 14 | S1 | 2020 | 10 | 23 | S2B |
| 2020 | 8 | 14 | S2B | 2020 | 10 | 25 | S1 |
| 2020 | 8 | 17 | S2B | 2020 | 10 | 26 | S2B |
| 2020 | 8 | 19 | S2A | 2020 | 10 | 28 | S2A |
| 2020 | 8 | 22 | S1 | 2020 | 10 | 31 | S2A |
| 2020 | 8 | 22 | S2A | 2020 | 11 | 2 | S1 |
| 2020 | 8 | 24 | S2B | 2020 | 11 | 2 | S2B |
| 2020 | 8 | 26 | S1 | 2020 | 11 | 5 | S2B |
| 2020 | 8 | 27 | S2B | 2020 | 11 | 6 | S1 |
| 2020 | 8 | 29 | S2A | 2020 | 11 | 7 | S2A |
| 2020 | 9 | 1 | S2A | 2020 | 11 | 10 | S2A |
| 2020 | 11 | 12 | S2B | 2021 | 1 | 24 | S2B |
| 2020 | 11 | 14 | S1 | 2021 | 1 | 25 | S1 |
| 2020 | 11 | 15 | S2B | 2021 | 1 | 26 | S2A |
| 2020 | 11 | 17 | S2A | 2021 | 1 | 29 | S1 |
| 2020 | 11 | 18 | S1 | 2021 | 1 | 29 | S2A |
| 2020 | 11 | 20 | S2A | 2021 | 1 | 31 | S2B |
| 2020 | 11 | 22 | S2B | 2021 | 2 | 3 | S2B |
| 2020 | 11 | 25 | S2B | 2021 | 2 | 5 | S2A |
| 2020 | 11 | 27 | S2A | 2021 | 2 | 6 | S1 |
| 2020 | 11 | 30 | S1 | 2021 | 2 | 8 | S2A |
| 2020 | 11 | 30 | S2A | 2021 | 2 | 10 | S1 |
| 2020 | 12 | 2 | S2B | 2021 | 2 | 10 | S2B |
| 2020 | 12 | 5 | S2B | 2021 | 2 | 13 | S2B |
| 2020 | 12 | 7 | S2A | 2021 | 2 | 15 | S2A |
| 2020 | 12 | 8 | S1 | 2021 | 2 | 18 | S1 |





| Year | Month | Day | Satellite | Year | Month | Day | Satellite |
|------|-------|-----|-----------|------|-------|-----|-----------|
| 2020 | 12 | 10 | S2A | 2021 | 2 | 18 | S2A |
| 2020 | 12 | 12 | S1 | 2021 | 2 | 20 | S2B |
| 2020 | 12 | 12 | S2B | 2021 | 2 | 22 | S1 |
| 2020 | 12 | 15 | S2B | 2021 | 2 | 23 | S2B |
| 2020 | 12 | 17 | S2A | 2021 | 2 | 25 | S2A |
| 2020 | 12 | 20 | S1 | 2021 | 2 | 28 | S2A |
| 2020 | 12 | 20 | S2A | 2021 | 3 | 2 | S2B |
| 2020 | 12 | 22 | S2B | 2021 | 3 | 5 | S2B |
| 2020 | 12 | 24 | S1 | 2021 | 3 | 6 | S1 |
| 2020 | 12 | 25 | S2B | 2021 | 3 | 7 | S2A |
| 2020 | 12 | 27 | S2A | 2021 | 3 | 10 | S2A |
| 2020 | 12 | 30 | S2A | 2021 | 3 | 12 | S2B |
| 2021 | 1 | 1 | S1 | 2021 | 3 | 14 | S1 |
| 2021 | 1 | 1 | S2B | 2021 | 3 | 15 | S2B |
| 2021 | 1 | 4 | S2B | 2021 | 3 | 17 | S2A |
| 2021 | 1 | 5 | S1 | 2021 | 3 | 18 | S1 |
| 2021 | 1 | 6 | S2A | 2021 | 3 | 20 | S2A |
| 2021 | 1 | 9 | S2A | 2021 | 3 | 22 | S2B |
| 2021 | 1 | 11 | S2B | 2021 | 3 | 25 | S2B |
| 2021 | 1 | 13 | S1 | 2021 | 3 | 26 | S1 |
| 2021 | 1 | 14 | S2B | 2021 | 3 | 27 | S2A |
| 2021 | 1 | 16 | S2A | 2021 | 3 | 30 | S1 |
| 2021 | 1 | 17 | S1 | 2021 | 3 | 30 | S2A |
| 2021 | 1 | 19 | S2A | 2021 | 4 | 1 | S2B |
| 2021 | 1 | 21 | S2B | 2021 | 4 | 4 | S2B |
| 2021 | 4 | 6 | S2A | 2021 | 6 | 18 | S2A |
| 2021 | 4 | 7 | S1 | 2021 | 6 | 20 | S2B |
| 2021 | 4 | 9 | S2A | 2021 | 6 | 23 | S2B |
| 2021 | 4 | 11 | S1 | 2021 | 6 | 25 | S2A |
| 2021 | 4 | 11 | S2B | 2021 | 6 | 28 | S2A |
| 2021 | 4 | 14 | S2B | 2021 | 6 | 30 | S1 |
| 2021 | 4 | 16 | S2A | 2021 | 6 | 30 | S2B |
| 2021 | 4 | 19 | S1 | 2021 | 7 | 3 | S2B |
| 2021 | 4 | 19 | S2A | 2021 | 7 | 5 | S2A |
| 2021 | 4 | 21 | S2B | 2021 | 7 | 8 | S2A |
| 2021 | 4 | 23 | S1 | 2021 | 7 | 10 | S2B |
| 2021 | 4 | 24 | S2B | 2021 | 7 | 12 | S1 |
| 2021 | 4 | 26 | S2A | 2021 | 7 | 13 | S2B |
| 2021 | 4 | 29 | S2A | 2021 | 7 | 15 | S2A |
| 2021 | 5 | 1 | S1 | 2021 | 7 | 16 | S1 |
| 2021 | 5 | 1 | S2B | 2021 | 7 | 18 | S2A |



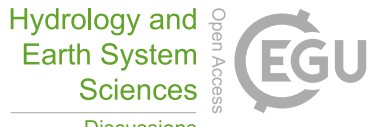

| Year | Month | Day | Satellite | Year | Month | Day | Satellite |
|------|-------|-----|-----------|------|-------|-----|-----------|
| 2021 | 5 | 4 | S2B | 2021 | 7 | 20 | S2B |
| 2021 | 5 | 5 | S1 | 2021 | 7 | 23 | S2B |
| 2021 | 5 | 6 | S2A | 2021 | 7 | 24 | S1 |
| 2021 | 5 | 9 | S2A | 2021 | 7 | 25 | S2A |
| 2021 | 5 | 11 | S2B | 2021 | 7 | 28 | S2A |
| 2021 | 5 | 13 | S1 | 2021 | 7 | 30 | S2B |
| 2021 | 5 | 14 | S2B | 2021 | 8 | 2 | S2B |
| 2021 | 5 | 16 | S2A | 2021 | 8 | 4 | S2A |
| 2021 | 5 | 17 | S1 | 2021 | 8 | 5 | S1 |
| 2021 | 5 | 19 | S2A | 2021 | 8 | 7 | S2A |
| 2021 | 5 | 21 | S2B | 2021 | 8 | 9 | S1 |
| 2021 | 5 | 24 | S2B | 2021 | 8 | 9 | S2B |
| 2021 | 5 | 25 | S1 | 2021 | 8 | 12 | S2B |
| 2021 | 5 | 26 | S2A | 2021 | 8 | 14 | S2A |
| 2021 | 5 | 29 | S2A | 2021 | 8 | 17 | S1 |
| 2021 | 5 | 31 | S2B | 2021 | 8 | 17 | S2A |
| 2021 | 6 | 3 | S2B | 2021 | 8 | 19 | S2B |
| 2021 | 6 | 5 | S2A | 2021 | 8 | 21 | S1 |
| 2021 | 6 | 6 | S1 | 2021 | 8 | 22 | S2B |
| 2021 | 6 | 8 | S2A | 2021 | 8 | 24 | S2A |
| 2021 | 6 | 10 | S2B | 2021 | 8 | 27 | S2A |
| 2021 | 6 | 13 | S2B | 2021 | 8 | 29 | S1 |
| 2021 | 6 | 15 | S2A | 2021 | 8 | 29 | S2B |
| 2021 | 6 | 18 | S1 | 2021 | 9 | 1 | S2B |
| 2021 | 9 | 2 | S1 | 2021 | 11 | 13 | S1 |
| 2021 | 9 | 3 | S2A | 2021 | 11 | 15 | S2A |
| 2021 | 9 | 6 | S2A | 2021 | 11 | 17 | S2B |
| 2021 | 9 | 8 | S2B | 2021 | 11 | 20 | S2B |
| 2021 | 9 | 10 | S1 | 2021 | 11 | 21 | S1 |
| 2021 | 9 | 11 | S2B | 2021 | 11 | 22 | S2A |
| 2021 | 9 | 13 | S2A | 2021 | 11 | 25 | S1 |
| 2021 | 9 | 14 | S1 | 2021 | 11 | 25 | S2A |
| 2021 | 9 | 16 | S2A | 2021 | 11 | 27 | S2B |
| 2021 | 9 | 18 | S2B | 2021 | 11 | 30 | S2B |
| 2021 | 9 | 21 | S2B | 2021 | 12 | 2 | S2A |
| 2021 | 9 | 22 | S1 | 2021 | 12 | 3 | S1 |
| 2021 | 9 | 23 | S2A | 2021 | 12 | 5 | S2A |
| 2021 | 9 | 26 | S1 | 2021 | 12 | 7 | S1 |
| 2021 | 9 | 26 | S2A | 2021 | 12 | 7 | S2B |
| 2021 | 9 | 28 | S2B | 2021 | 12 | 10 | S2B |
| 2021 | 10 | 1 | S2B | 2021 | 12 | 12 | S2A |



| Year | Month | Day | Satellite | Year | Month | Day | Satellite |
|------|-------|-----|-----------|------|-------|-----|-----------|
| 2021 | 10 | 3 | S2A | 2021 | 12 | 15 | S1 |
| 2021 | 10 | 4 | S1 | 2021 | 12 | 15 | S2A |
| 2021 | 10 | 6 | S2A | 2021 | 12 | 17 | S2B |
| 2021 | 10 | 8 | S1 | 2021 | 12 | 19 | S1 |
| 2021 | 10 | 8 | S2B | 2021 | 12 | 20 | S2B |
| 2021 | 10 | 11 | S2B | 2021 | 12 | 22 | S2A |
| 2021 | 10 | 13 | S2A | 2021 | 12 | 25 | S2A |
| 2021 | 10 | 16 | S1 | 2021 | 12 | 27 | S1 |
| 2021 | 10 | 16 | S2A | 2021 | 12 | 27 | S2B |
| 2021 | 10 | 18 | S2B | 2021 | 12 | 30 | S2B |
| 2021 | 10 | 20 | S1 | 2021 | 12 | 31 | S1 |
| 2021 | 10 | 21 | S2B | *Remark | | | |
| 2021 | 10 | 23 | S2A | S1   = Sentinel – 1 | | | |
| 2021 | 10 | 26 | S2A | S2A = Sentinel – 2A | | | |
| 2021 | 10 | 28 | S1 | S2B = Sentinel – 2B | | | |
| 2021 | 10 | 28 | S2B | | | | |
| 2021 | 10 | 31 | S2B | | | | |
| 2021 | 11 | 1 | S1 | | | | |
| 2021 | 11 | 2 | S2A | | | | |
| 2021 | 11 | 5 | S2A | | | | |
| 2021 | 11 | 7 | S2B | | | | |
| 2021 | 11 | 9 | S1 | | | | |
| 2021 | 11 | 10 | S2B | | | | |

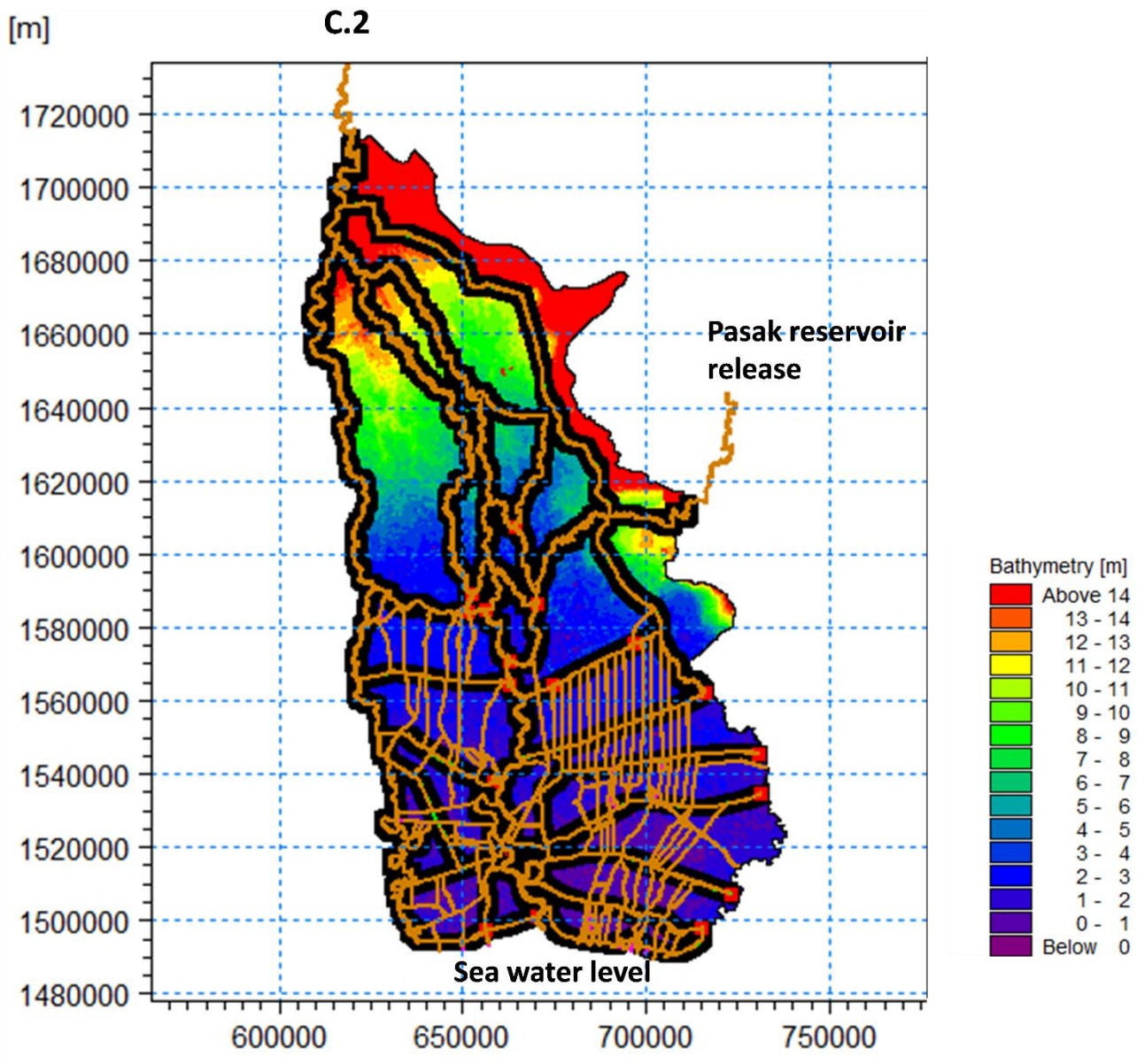

**Figure A 1: 1D-2D Flood model**


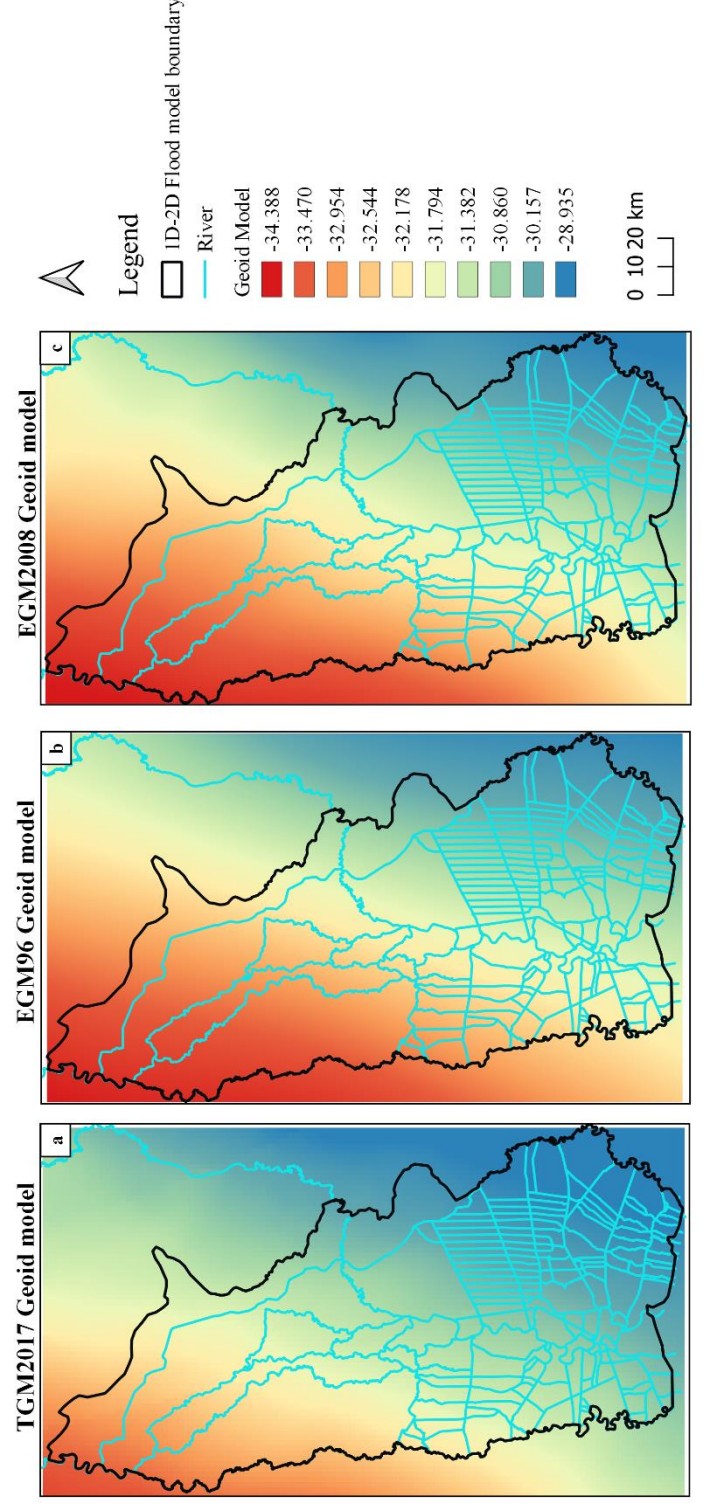

**Figure A 2: The geoid model dataset (a) TGM2017, (b) EGM96 and (c) EGM2008**





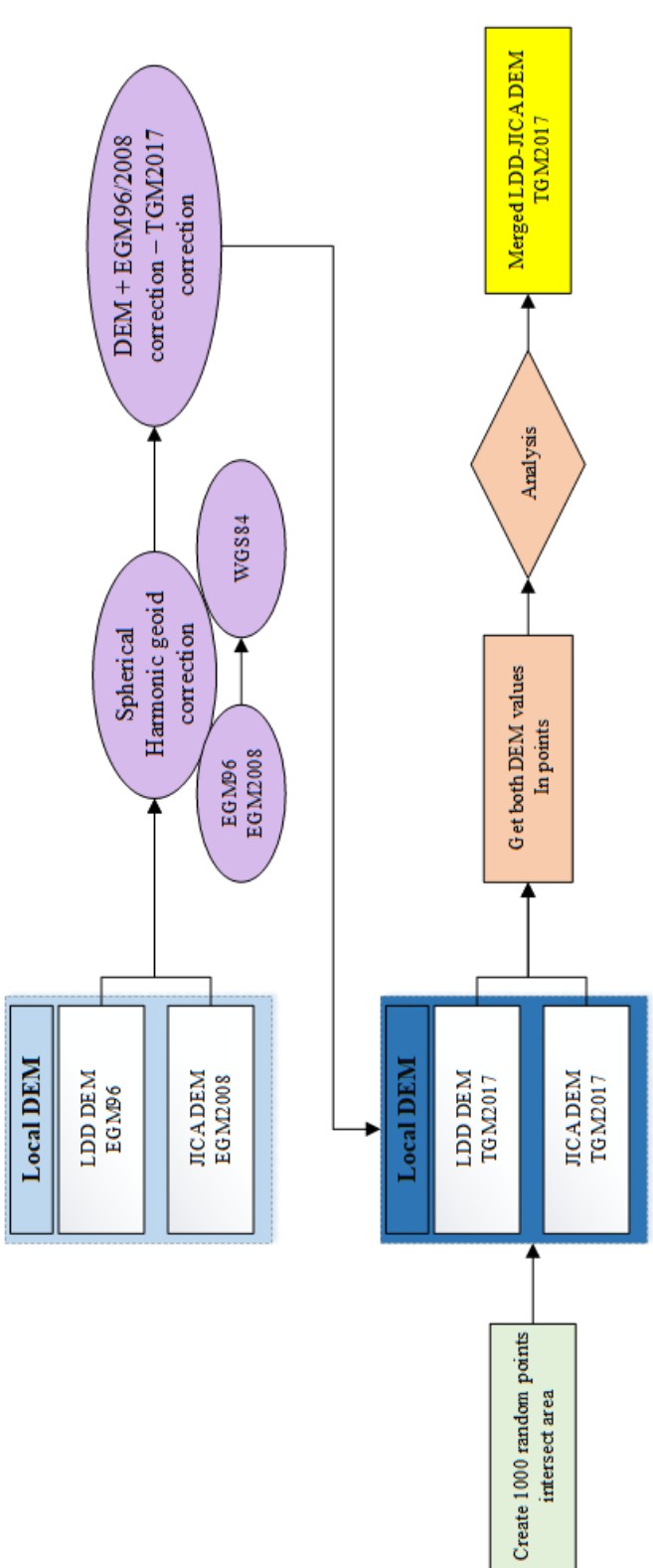

**Figure A 3: the merged LDD-JICA DEM's workflow**



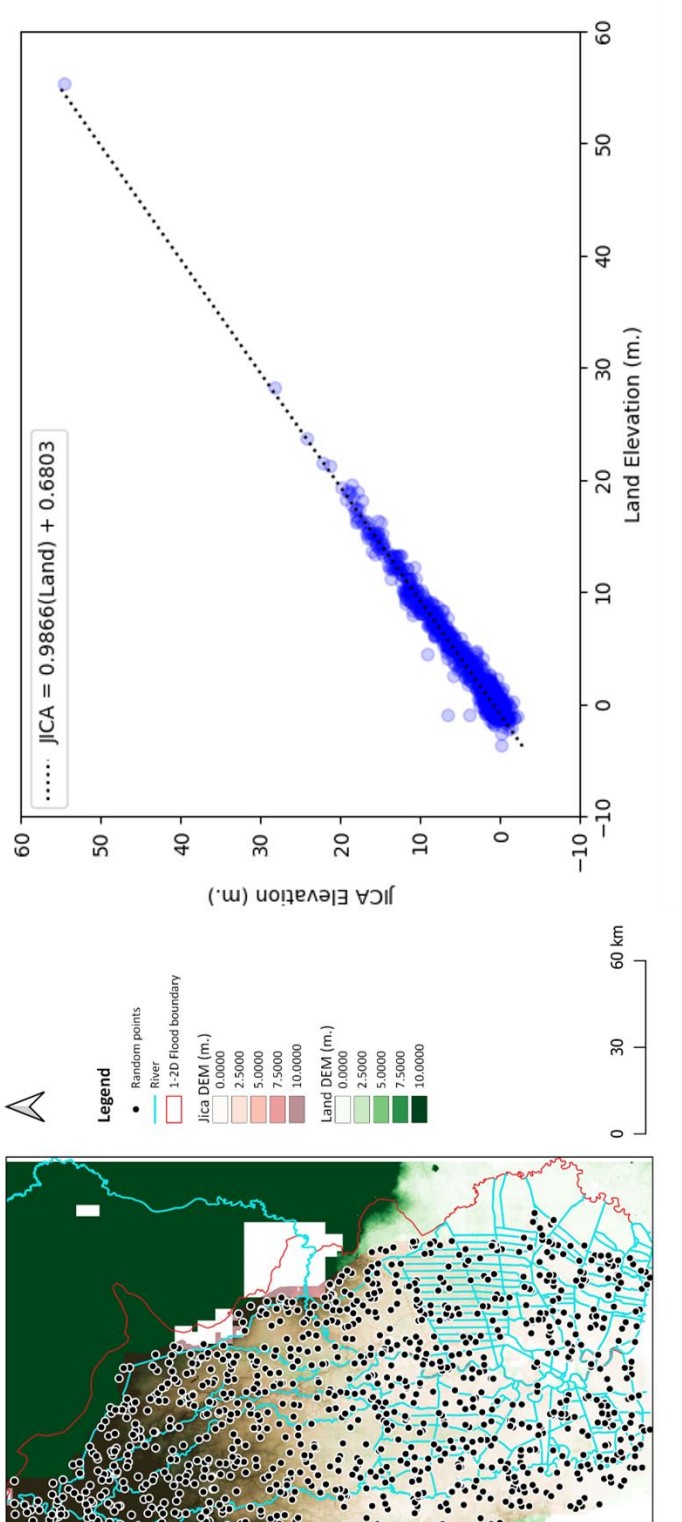

**Figure A 4: the merged LDD-JICA DEM's processing**

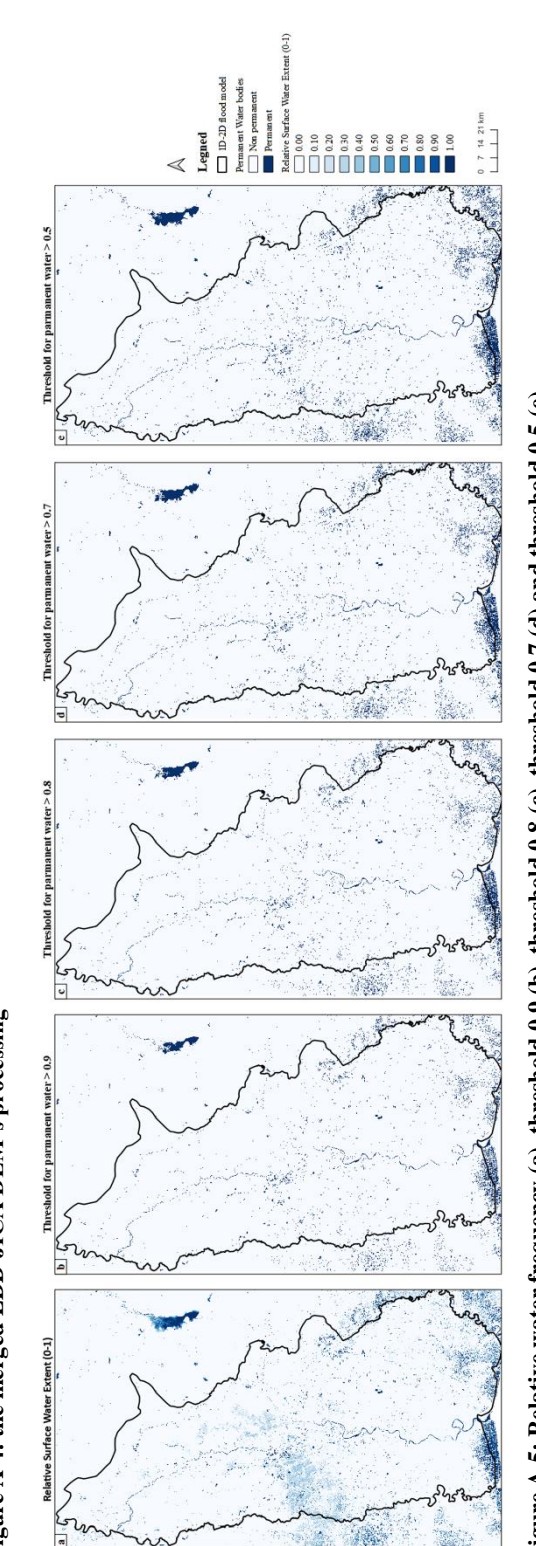

**Figure A 5: Relative water frequency (a), threshold 0.9 (b), threshold 0.8 (c), threshold 0.7 (d) and threshold 0.5 (e).**





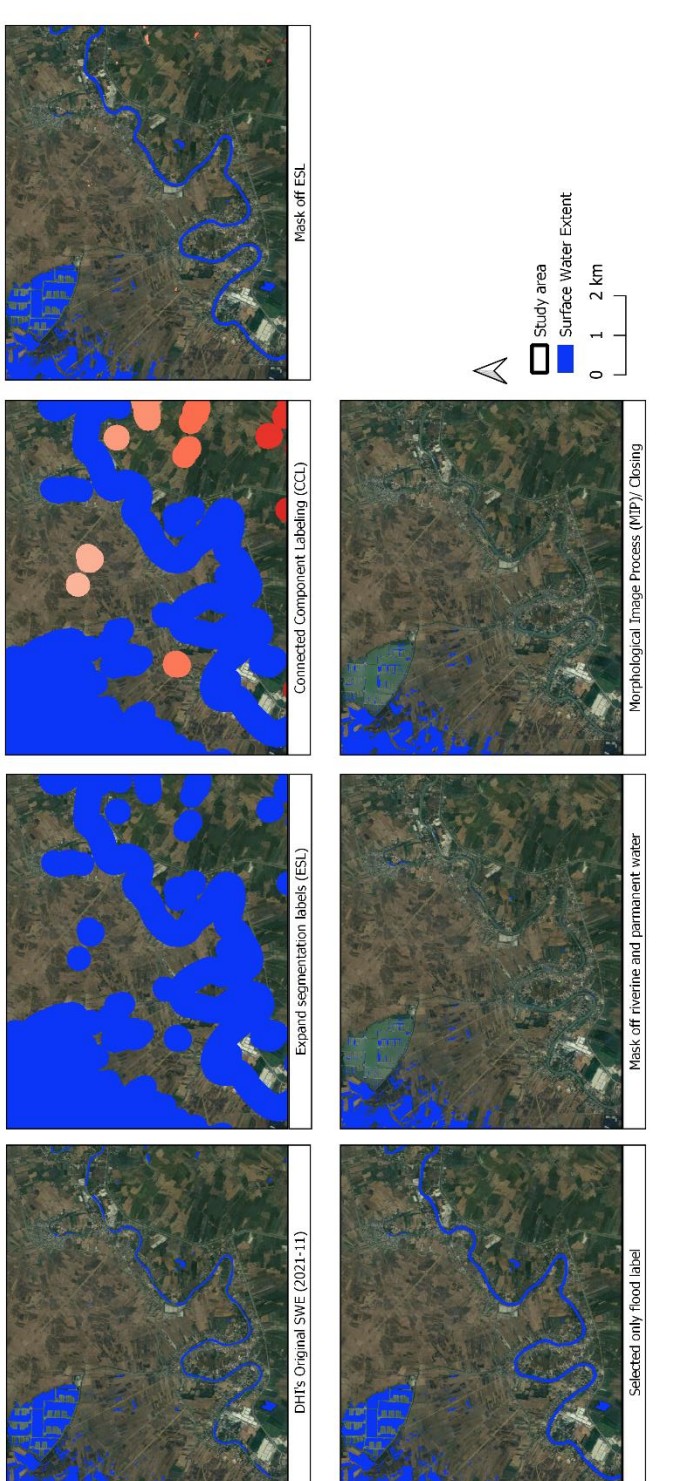

**Figure A 6: Riverine flood classification processing. © Google Earth**

**Figure A 7: The histogram distribution of the mean error (ME), comparing 10 DEM products against the ICESat-2 ATL07 benchmark.**





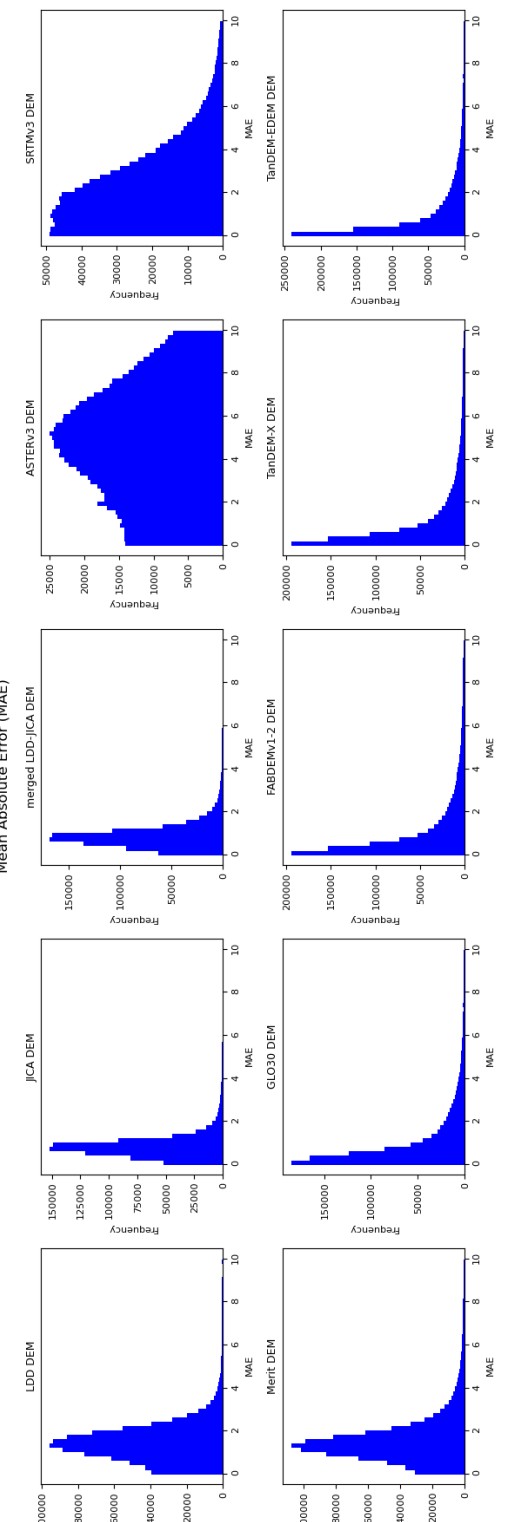

**Figure A 8: The histogram distribution of the mean error (MAE), comparing 10 DEM products against the ICESat-2 ATL07 benchmark.**

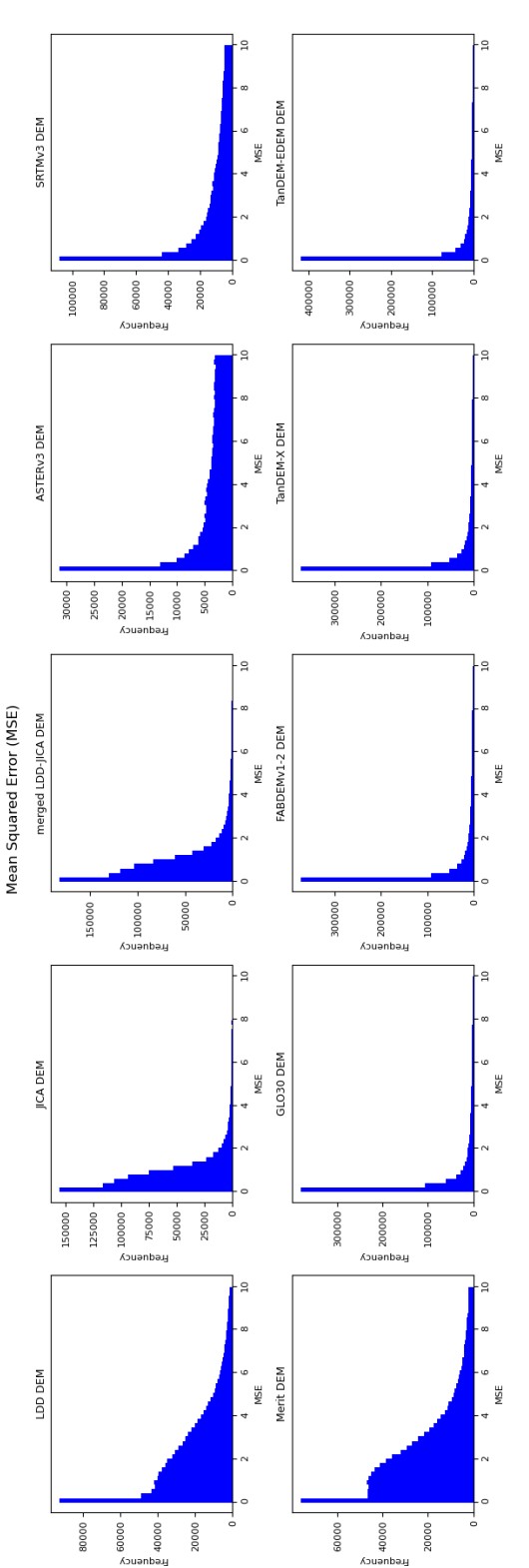

**Figure A 9: The histogram distribution of the mean error (MSE), comparing 10 DEM products against the ICESat-2 ATL07 benchmark.**





**Figure A 10: The mean error (ME) spatial grid comparison of 10 DEM products against the ICESat-2 ATL08 benchmark, with a resolution of 5x5 km.**





**Figure A 11: The mean error (MAE) spatial grid comparison of 10 DEM products against the ICESat-2 ATL08 benchmark, with a resolution of 5x5 km.**

**Figure A 12: The mean error (RMSE) spatial grid comparison of 10 DEM products against the ICESat-2 ATL08 benchmark, with a resolution of 5x5 km.**

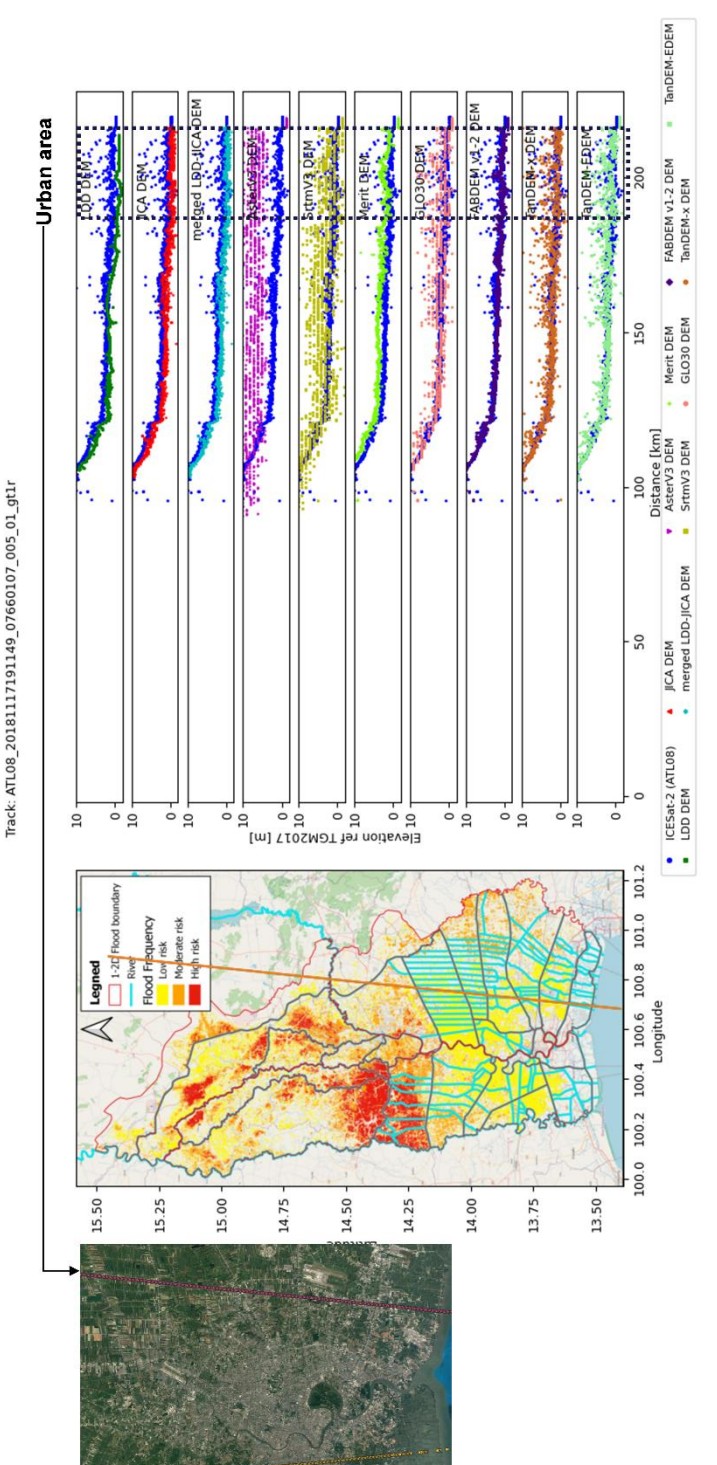

**Figure A 13: The track-wise comparison of 10 DEM products with ICESat-2 ATL08 benchmark. © OpenStreetMap contributors 2015. Distributed under the Open Data Commons Open Database License (ODbL) v1.0. © Google Earth.**



**Author contributions**

Theerapol Charoensuk: Conceptualization, Methodology, Validation, Formal analysis, Investigation, Data curation, Visualization, Writing – review & editing, Writing – original draft. Claudia Katrine Corvenius Lorentzen: Coding and Visualization. Anne Beukel Bak: Coding and Visualization. Jakob Luchner: Software, review. Christian Tøttrup: Data curation, review. Peter Bauer-Gottwein: Methodology, Conceptualization, Resources, Supervision, Writing – review & editing.

**Competing interests**

The authors declare that they have no conflict of interest.

**Acknowledgements**

We gratefully acknowledge the Hydro-Informatics Institute (HII), DHI A/S, the Geo-Informatics and Space Technology Development Agency (GISTDA), German Aerospace Centre (DLR) and the WorldWater project (https://worldwater.earth/) 725 funded by the European Commission and European Space Agency (ESA) for providing historical observed data, in-situ data, Chao Phraya's 1D-2D flood models, MIKE powered by DHI software, flood maps, digital elevation model and water surface extent data. This study received no funding. Theerapol Charoensuk received financial support from the Office of the Civil Service Commission (OCSC) scholarship provided by the Thai Royal government. Their unwavering support and invaluable contributions have been instrumental in the fruition of this work.

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
