# Peer review of "Enhancing the performance of 1D-2D flood models using satellite laser altimetry and multi-mission surface water extent maps from Earth Observation (EO) data"

_Hydrology and Earth System Sciences, 2024_

## Author Comment (AC1)

We sincerely thank the reviewers for their valuable comments and suggestions, which have significantly contributed to improving the manuscript. Below, we have reproduced the reviewers' comments in black font, followed by our responses in blue font.

1. The author must constructively change the abstract in terms of adding error analysis values in terms of **PBIAS** to the result. The Author needs to write consistently.

In the abstract, we present the DEM analysis using root mean square error (RMSE) and will revise it to include the PBIAS results. Additionally, we will incorporate the PBIAS equation into Section 4.1.3, Evaluation of DEMs using the ICESat-2 ATL08 Benchmark, and add the PBIAS results to Table 6, which summarizes the statistical metrics comparing 10 DEM products against the ICESat-2 benchmark.

$$PBIAS = 100 \, x \left[ \frac{\sum_{i=1}^{n}(Y_i - \hat{Y}_i)}{\sum_{i-1}^{n} \hat{Y}_i} \right]$$

Where $\hat{Y}i$ represents ICESat-2 ATL08 elevation, $Y_i$ denotes the elevation for each DEM (i.e., LDD DEM, JICA, merged LDD-JICA DEM, ASTEM GDEM V3, SRTM DEM, MERIT DEM, FABDEM v1-2 DEM, GLO30 DEM, TanDEM-X, and TanDEM-EDEM), and $n$ is the number of observations.

The ideal value of PBIAS is 0: positive values indicate that the DEM products are biased toward overestimating compared to the ICESat-2 ATL08 benchmark, while negative values indicate a bias toward underestimation.

Table 6: Table of statistical metrics, comparing 10 DEM products against the ICESat-2 benchmark. The resulting averages are computed across the datasets in study area.

| DEM product | Statistical method | | | | |
|---|---|---|---|---|---|
| | ME (m.) | MAE (m.) | MSE (m.) | RMSE (m.) | PBIAS (-) |
| LDD | -1.3 | 1.64 | 5.45 | 2.33 | -34.76 |
| JICA | -0.65 | 1.04 | 3.51 | 1.87 | -17.00 |
| merged LDD-JICA | -0.68 | 1.08 | 3.74 | 1.93 | -15.38 |
| ASTER | 4.77 | 5.57 | 44.28 | 6.65 | 47.71 |
| SRTM | 2.04 | 2.58 | 12.92 | 3.59 | 27.99 |
| MERIT | 1.56 | 1.79 | 6.76 | 2.6 | 22.99 |
| GlO30 | 0.84 | 1.3 | 5.89 | 2.43 | 13.87 |
| FABDEMv1-2 | 0.25 | 0.8 | 3.79 | 1.95 | 4.59 |
| TanDEM-X | 0.94 | 1.73 | 13.29 | 3.65 | 15.24 |
| TanDEM-EDEM | 0.91 | 1.43 | 7.74 | 2.78 | 14.84 |

2. The author must mention where they got data and the frequency of data. Statistical analysis of data must be given in Tabular format (Like Table no 1, 10.1061/(ASCE)IR.1943-4774.0001689).

Regarding the reference from Table 1 (10.1061/(ASCE)IR.1943-4774.0001689), it provides statistics on runoff data.

**Table 1.** Descriptive statistics of runoff data for 0° slope

| Statistical parameters | Training set (252) | Testing set (108) | Total data set (360) |
|---|---|---|---|
| | Rainfall intensity 1 L/min | | |
| Min | 0.01 | 0.03 | 0.01 |
| Max | 1.07 | 1.07 | 1.07 |
| Mean | 0.511 | 0.593 | 0.536 |
| Kurtosis | −0.763 | −0.874 | −0.842 |
| Skewness | 0.211 | −0.090 | 0.127 |
| SD | 0.283 | 0.297 | 0.289 |

However, this paper focuses on DEM products and the impact of the DEM on hydraulic modeling, as explained in Table 2. We will include a detailed statistical analysis of the DEM products, which will be added in the appendix, as shown in the table below.

| DEM product | Statistical Parameters | | | | |
|---|---|---|---|---|---|
| | Min | Max | Mean | Standard Deviation | Median |
| ICESat-2 ATL08 | -7.00 | 218.42 | 5.29 | 6.81 | 2.49 |
| LDD | -9.41 | 254.27 | 4.34 | 7.75 | 1.51 |
| JICA | -22.97 | 239.31 | 4.20 | 5.48 | 1.95 |
| merged LDD-JICA | -16.00 | 378.73 | 5.21 | 8.26 | 1.87 |
| ASTER | -2.00 | 267.93 | 6.23 | 8.23 | 2.85 |
| SRTM | -34.97 | 262.17 | 8.02 | 8.47 | 5.25 |
| MERIT | -1.29 | 257.32 | 7.53 | 8.17 | 4.34 |
| GlO30 | -15.93 | 271.15 | 6.87 | 8.30 | 4.15 |
| FABDEMv1-2 | -14.99 | 267.93 | 6.22 | 8.23 | 2.85 |
| TanDEM-X | -7.00 | 274.93 | 7.06 | 8.48 | 4.24 |
| TanDEM-EDEM | -36.91 | 271.26 | 6.93 | 8.35 | 4.13 |

3. Fig 2: it should be clearly described in terms of scientific manner.

We will revise and incorporate this description into the manuscript, as illustrated in the figure below:

[Figure]

Figure 2: ICESat-2 ATL08 and DEM products, including: A) ICESat-2 ATL08 surface elevation, B) Land Development Department (LDD) DEM, C) JICA DEM, D) Merged LDD.JICA DEM, E) ASTER GDEM Version 3, F) SRTM DEM Version 3, G) MERIT DEM, H) GLO-30 DEM, I) FABDEM v1.2 DEM, J) TanDEM-X DEM, and K) TanDEM-EDM.

4. For better understanding, Please add more recent literature (2024) regarding the bidirectional flood models using satellite laser altimetry.

Although recent literature from 2024 is limited, we will revise the manuscript to include references from Nandam and Patel, 2024, which evaluated the suitability of global DEMs for hydrodynamic modeling in data-scarce regions using satellite laser altimetry, and Frias et al., 2024, which enhanced the accuracy of 2D hydraulic models through DEM correction using machine learning and satellite laser altimetry. These studies will be cited in the introduction.

5. Please modify the objective section for a clear understanding, i.e., the novelty part should be mentioned.

We will revise the manuscript. We believe that our paper has two novel aspects: (1) Comprehensive DEM evaluation against ICESat-2 benchmark for the Thailand domain. (2) Systematic comparison of simulated 2D inundation patterns with inundation patterns derived from satellite EO flooding patterns.

6. There are so many techniques in the recent world for the flood model; why does the author use a specified Method for research purposes? Is there any specific reason for this?

Currently, numerous techniques exist to enhance the performance of 1D-2D flood models. However, the accuracy of simulated floods heavily depends on the quality of the DEM data, making it essential to validate DEMs before use. Land use is constantly changing, and surveying DEMs is both time-consuming and expensive. Global DEM products offer a viable alternative, as they are often freely

available and up-to-date, but they still require validation prior to implementation. ICESat-2, which continues to operate and measure surface elevation, provides valuable data for validating DEMs.

Riverine classification based on surface water extent (SWE) from satellite data remains a significant challenge with limited literature available. In this study, we applied new techniques to address the issue.

7. The author must add statistical components/parameters of collected data in the case study section.

As explained in response to Question 2, we will include this information in the appendix.

8. Eq 3-6; please add a recent citation for reference purposes. [Read this paper: 10.1016/j.gsd.2024.101178, 10.1016/j.clwat.2024.100003, 10.1016/j.hydres.2024.04.006, 10.1007/978-981-15-5397-4_75, 10.1038/s41598-024-63490-1, 10.2166/wcc.2021.221]

References in the equations will be added.

9. A comparison statement (compare with other research articles) must be added in the result and discussion section to visualize the proposed research better.

We will revise the manuscript accordingly; however, the discussion and comparison with other research articles are already addressed in Section 6.1, Overall Results of DEM Analysis Workflow, where the results are compared with findings from other studies

10. The author must add future scope in the last portion of the manuscript.

We will include a new section, Future Applications, in the manuscript to explore potential advancements and areas for further research.

11. The advantages and limitations of the proposed model must be added.

We will revise the manuscript, particularly the Discussion section, to thoroughly address both the advantages and limitations of the study.

12. For better analysis of the result, the author must add a Box plot, Taylor diagram, and ROC Curve

We will revise the manuscript add box plot in section 5.3 Results of the evaluation of flood inundation maps, as illustrated in the figure below:

[Figure]

13. The author must provide a flow chart and parameter table of proposed individual models.

We have explained the overall workflow in the manuscript, as shown in Figure 4, and we will revise it to enhance clarity and ensure it better aligns with the workflow process.

[Figure]

Figure 4: Overall Methodology: (a) Component 1: DEM Analysis – Involves processing ICESat-2 ATL08 data, applying vertical datum referencing, and evaluating DEMs against the ICESat-2 ATL08 benchmark through point, grid, and track-wise comparisons.  (b) Component 2: Flood Map Analysis – Includes setting up the 1D-2D flood model, performing flood classification, and evaluating flood maps using appropriate methods.

14. The author considered different input constraints; is there any scientific reason for the same

For the DEM analysis, we selected a range of DEM products based on their resolution and data acquisition methods to identify the most suitable for the study area. The reasoning behind this is that DEM accuracy plays a crucial role in flood modeling, as elevation data influences water flow simulations and flood inundation predictions. We chose two DEMs for in-depth analysis: one from a local DEM survey conducted by a Thai agency, providing region-specific accuracy, and a global DEM derived from satellite data, offering broader coverage. By evaluating the flood maps generated from these different DEM sources, we aimed to determine which DEM provided the most reliable and accurate representation of the terrain and flood patterns in the study area, thus ensuring that the hydrodynamic models produced the most realistic flood inundation simulations.

**Reference**

Frias, M.C., Liu, S., Mo, X., Druce, D., Yamazaki, D., Folkmann, A., Nielsen, K., Bauer-gottwein, P., 2024. Improving 2D hydraulic modeling in floodplain areas with ICESat-2 data : A case study in Upstream Yellow River. EGU General Assembly 2024, Vienna, Austria, 14–19 Apr 2024, EGU24-14669, pp. 24–25. doi:https://doi.org/10.5194/egusphere-egu24-14669

Nandam, V., Patel, P.L., 2024. A framework to assess suitability of global digital elevation models for hydrodynamic modelling in data scarce regions. J. Hydrol. 630, 130654. doi:10.1016/j.jhydrol.2024.130654

---

## Author Response (AR1)

We sincerely thank the reviewers for their valuable comments and suggestions, which have significantly contributed to improving the manuscript. Below, we have reproduced the reviewers' comments in black font, followed by our responses in blue font.

**Reviewer 1**

1. The author must constructively change the abstract in terms of adding error analysis values in terms of **PBIAS** to the result. The Author needs to write consistently.

In the abstract, we present the DEM analysis using root mean square error (RMSE) and have revised it to include the PBIAS results. Additionally, we have incorporated the PBIAS equation into Section 4.1.3, Evaluation of DEMs using the ICESat-2 ATL08 Benchmark, and add the PBIAS results to Table 6, which summarizes the statistical metrics comparing 10 DEM products against the ICESat-2 benchmark.

$$PBIAS = 100 \, x \left[ \frac{\sum_{i=1}^{n}(Y_i - \hat{Y}_i)}{\sum_{i-1}^{n} \hat{Y}_i} \right]$$

Where $\hat{Y}i$ represents ICESat-2 ATL08 elevation, $Yi$ denotes the elevation for each DEM (i.e., LDD DEM, JICA, merged LDD-JICA DEM, ASTEM GDEM V3, SRTM DEM, MERIT DEM, FABDEM v1-2 DEM, GLO30 DEM, TanDEM-X, and TanDEM-EDEM), and $n$ is the number of observations.

The ideal value of PBIAS is 0: positive values indicate that the DEM products are biased toward overestimating compared to the ICESat-2 ATL08 benchmark, while negative values indicate a bias toward underestimation.

Table 6: Table of statistical metrics, comparing 10 DEM products against the ICESat-2 benchmark. The resulting averages are computed across the datasets in study area.

| DEM product | Statistical method | | | | |
|---|---|---|---|---|---|
| | ME (m.) | MAE (m.) | MSE (m.) | RMSE (m.) | PBIAS (%) |
| LDD | -1.3 | 1.64 | 5.45 | 2.33 | -34.76 |
| JICA | -0.65 | 1.04 | 3.51 | 1.87 | -17.00 |
| merged LDD-JICA | -0.68 | 1.08 | 3.74 | 1.93 | -15.38 |
| ASTER | 4.77 | 5.57 | 44.28 | 6.65 | 47.71 |
| SRTM | 2.04 | 2.58 | 12.92 | 3.59 | 27.99 |
| MERIT | 1.56 | 1.79 | 6.76 | 2.6 | 22.99 |
| GlO30 | 0.84 | 1.3 | 5.89 | 2.43 | 13.87 |
| FABDEMv1-2 | 0.25 | 0.8 | 3.79 | 1.95 | 4.59 |
| TanDEM-X | 0.94 | 1.73 | 13.29 | 3.65 | 15.24 |
| TanDEM-EDEM | 0.91 | 1.43 | 7.74 | 2.78 | 14.84 |

**We have revised the abstract (Lines 20–22), Section 4.1.3: Evaluation of DEMs Using ICESat-2 ATL08 Benchmark (Lines 358–370), and updated Table 6 in Section 5.2.1: Point Comparison Evaluation Results (Lines 479–451 and 490–494) of the revised manuscript.**

2. The author must mention where they got data and the frequency of data. Statistical analysis of data must be given in Tabular format (Like Table no 1, 10.1061/(ASCE)IR.1943-4774.0001689).

Regarding the reference from Table 1 (10.1061/(ASCE)IR.1943-4774.0001689), it provides statistics on runoff data.

**Table 1.** Descriptive statistics of runoff data for 0° slope

| Statistical parameters | Training set (252) | Testing set (108) | Total data set (360) |
|---|---|---|---|
| | Rainfall intensity 1 L/min | | |
| Min | 0.01 | 0.03 | 0.01 |
| Max | 1.07 | 1.07 | 1.07 |
| Mean | 0.511 | 0.593 | 0.536 |
| Kurtosis | −0.763 | −0.874 | −0.842 |
| Skewness | 0.211 | −0.090 | 0.127 |
| SD | 0.283 | 0.297 | 0.289 |

However, this paper focuses on DEM products and the impact of the DEM on hydraulic modeling, as explained in Table 2. We have included a detailed statistical analysis of the DEM products, which was added in the appendix, as shown in the table below.

| DEM product | Statistical Parameters | | | | |
|---|---|---|---|---|---|
| | Min | Max | Mean | Standard Deviation | Median |
| ICESat-2 ATL08 | -7.00 | 218.42 | 5.29 | 6.81 | 2.49 |
| LDD | -9.41 | 254.27 | 4.34 | 7.75 | 1.51 |
| JICA | -22.97 | 239.31 | 4.20 | 5.48 | 1.95 |
| merged LDD-JICA | -16.00 | 378.73 | 5.21 | 8.26 | 1.87 |
| ASTER | -2.00 | 267.93 | 6.23 | 8.23 | 2.85 |
| SRTM | -34.97 | 262.17 | 8.02 | 8.47 | 5.25 |
| MERIT | -1.29 | 257.32 | 7.53 | 8.17 | 4.34 |
| GlO30 | -15.93 | 271.15 | 6.87 | 8.30 | 4.15 |
| FABDEMv1-2 | -14.99 | 267.93 | 6.22 | 8.23 | 2.85 |
| TanDEM-X | -7.00 | 274.93 | 7.06 | 8.48 | 4.24 |
| TanDEM-EDEM | -36.91 | 271.26 | 6.93 | 8.35 | 4.13 |

**We have revised Section 3.3 Digital Elevation Models (DEM) (Lines 159), and updated Table A 1: Descriptive statistics of ten different DEM products in Appendix A of the revised manuscript.**

3. Fig 2: it should be clearly described in terms of scientific manner.

We have revised and incorporated this description into the manuscript, as illustrated in the figure below:

[Figure]

Figure 2: ICESat-2 ATL08 and DEM products, including: A) ICESat-2 ATL08 surface elevation, B) Land Development Department (LDD) DEM, C) JICA DEM, D) Merged LDD.JICA DEM, E) ASTER GDEM Version 3, F) SRTM DEM Version 3, G) MERIT DEM, H) GLO-30 DEM, I) FABDEM v1.2 DEM, J) TanDEM-X DEM, and K) TanDEM-EDM.

**We have updated the description and Figure 2 on Line 169, Page 8, of the revised manuscript.**

4. For better understanding, Please add more recent literature (2024) regarding the bidirectional flood models using satellite laser altimetry.

Although recent literature from 2024 is limited, we have revised the manuscript to include references from Nandam and Patel, 2024, which evaluated the suitability of global DEMs for hydrodynamic modeling in data-scarce regions using satellite laser altimetry, and Frias et al., 2024, which enhanced the accuracy of 2D hydraulic models through DEM correction using machine learning and satellite laser altimetry. These studies will be cited in the introduction.

**We have revised on lines 64 – 65 in Section 1: introduction of the revised manuscript.**

5. Please modify the objective section for a clear understanding, i.e., the novelty part should be mentioned.

We have revised the manuscript. We believe that our paper has two novel aspects: (1) Comprehensive DEM evaluation against ICESat-2 benchmark for the Thailand domain. (2) Systematic comparison of simulated 2D inundation patterns with inundation patterns derived from satellite EO flooding patterns.

**We have revised on lines 86-96 in Section 1: introduction of the revised manuscript.**

6. There are so many techniques in the recent world for the flood model; why does the author use a specified Method for research purposes? Is there any specific reason for this?

Currently, numerous techniques exist to enhance the performance of 1D-2D flood models. However, the accuracy of simulated floods heavily depends on the quality of the DEM data, making it essential to validate DEMs before use. Land use is constantly changing, and surveying DEMs is both timeconsuming and expensive. Global DEM products offer a viable alternative, as they are often freely available and up-to-date, but they still require validation prior to implementation. ICESat-2, which continues to operate and measure surface elevation, provides valuable data for validating DEMs.

Riverine classification based on surface water extent (SWE) from satellite data remains a significant challenge with limited literature available. In this study, we applied new techniques to address the issue.

**No changes have been implemented in the manuscript in response to this comment.**

7. The author must add statistical components/parameters of collected data in the case study section.

As explained in response to Question 2, we have included this information in the appendix.

**We have revised Section 3.3 Digital Elevation Models (DEM) (Lines 159), and updated Table A 1: Descriptive statistics of ten different DEM products in Appendix A of the revised manuscript.**

8. Eq 3-6; please add a recent citation for reference purposes. [Read this paper: 10.1016/j.gsd.2024.101178, 10.1016/j.clwat.2024.100003, 10.1016/j.hydres.2024.04.006, 10.1007/978-981-15-5397-4_75, 10.1038/s41598-024-63490-1, 10.2166/wcc.2021.221]

**We have revised in Section: 4.1.3 Evaluation of DEMs using ICESat-2 ATL08 Benchmark on lines 359-360 of the revised manuscript.**

9. A comparison statement (compare with other research articles) must be added in the result and discussion section to visualize the proposed research better.

We have revised the manuscript accordingly; however, the discussion and comparison with other research articles are already addressed in Section 6.1, Overall Results of DEM Analysis Workflow, where the results are compared with findings from other studies

**We have revised in Section 6.1 Overall result of DEM analysis workflow on lines 658-664 of the revised manuscript**

10. The author must add future scope in the last portion of the manuscript.

**We have updated new Section 6.4 : Future Applications on lines 720-236 of the revised manuscript.**

11. The advantages and limitations of the proposed model must be added.

We have revised the manuscript, particularly the Discussion section, to thoroughly address both the advantages and limitations of the study.

**We have updated new Section 6.3 Advantages and Limitations on lines 702-722 of the revised manuscript.**

12. For better analysis of the result, the author must add a Box plot, Taylor diagram, and ROC Curve

We have revised the manuscript add box plot in section 5.3 Results of the evaluation of flood inundation maps, as illustrated in the figure below:

[Figure]

**We have revised Section 5.3 Results of the evaluation of flood inundation maps on lines 621-632 and updated the Figure 12: box plots illustrating the performance of the flood model based on the merged LDD-JICA and FABDEMv1-2 DEMs across three statistical metrics: (a) Probability of Detection (POD), (b) False Alarm Ratio (FAR), and (c) Critical Success Index (CSI) on line 645 of the revised manuscripts.**

13. e author must provide a flow chart and parameter table of proposed individual models.

We have explained the overall workflow in the manuscript, as shown in Figure 4, and we have revised it to enhance clarity and ensure it better aligns with the workflow process.

[Figure]

Figure 4: Overall Methodology: (a) Component 1: DEM Analysis – Involves processing ICESat-2 ATL08 data, applying vertical datum referencing, and evaluating DEMs against the ICESat-2 ATL08 benchmark through point, grid, and track-wise comparisons. (b) Component 2: Flood Map Analysis – Includes setting up the 1D-2D flood model, performing flood classification, and evaluating flood maps using appropriate methods.

**We have updated the description and Figure 4 on Line 340, Page 16, of the revised manuscript.**

14. The author considered different input constraints; is there any scientific reason for the same

For the DEM analysis, we selected a range of DEM products based on their resolution and data acquisition methods to identify the most suitable for the study area. The reasoning behind this is that DEM accuracy plays a crucial role in flood modeling, as elevation data influences water flow simulations and flood inundation predictions. We chose two DEMs for in-depth analysis: one from a local DEM survey conducted by a Thai agency, providing region-specific accuracy, and a global DEM derived from satellite data, offering broader coverage. By evaluating the flood maps generated from these different DEM sources, we aimed to determine which DEM provided the most reliable and accurate representation of the terrain and flood patterns in the study area, thus ensuring that the hydrodynamic models produced the most realistic flood inundation simulations.

**No changes to the manuscript were implemented in response to this comment.**

**Reviewer 2**

General comments

Interesting study on the use of different DEM inputs into a specific 1D-2D flood model. In the first part, the most accurate global and airborne DEMs are determined. Using these two DEMs as input, the resulting flood model maps are compared with 2 reference flood maps. Surprisingly, the global flood-optimized FABDEM derived from TanDEM-X achieves only slightly worse flood modelling quality statistics than the higher-resolution airborne DEM version LDD-JICA DEM.

The paper is generally well structured and balanced, but the context and objective of the two parts are not consistently clear.

The claimed objective of your study to present two new workflows for updating 1D-2D flood models is not plausible. It seems like your starting point for this study was "what can we do with EO data", but what you describe is how to test the input DEMs and validate your results. I can't see any real improvement in the 1D-2D flood model itself:An extensive DEM evaluation does not make sense for every new 1D-2D flood model, as the number of input DEMs is limited and you have shown extensive evaluation here. Similarly, the validation of flood modeling results with existing flood maps is not an integral update of a model.

In my opinion, there is a simple way out: move the "workflows" to the discussion/ conclusion and stick to terms like "evaluation of DEMs for" … and "validation of flood model results" … .

However, please clarify this throughout the paper, even in the title!!!

The objective of this study is to enhance the performance of a 1D-2D flood model using Earth Observation (EO) data, with a focus on improving the accuracy of flood inundation simulations through Digital Elevation Model (DEM) analysis. We aim to refine the approach to 1D-2D flood modeling by integrating EO data into two key workflows: DEM analysis and flood map analysis. These workflows are designed to produce more accurate flood model results. Accordingly, we will change the title from "Upgrading 1D-2D flood models using satellite laser altimetry and multi-mission satellite surface water extent maps" to "Enhancing the performance of 1D-2D flood models using satellite laser altimetry and multi-mission surface water extent maps from Earth Observation (EO) data."'

**We have revised the title on lines 1 – 3.**

Specific comments

1. Please improve the abstract (and title) with regard to the readability and research focus of your study. Main point: The paper gives a kind of performance test. So, your description given in 4.2. (" .. to evaluate the performance of simulated flood maps … using various DEM products") comprises the content more appropriate than an "upgrade by two workflows".

As explained in response to General comments, we have revised this.

**We have revised the title and the abstract on lines 1 – 31.**

2. In that sense, unclear in the abstract: are you evaluating the 1D-2D model results with the surface water extent maps or has this any relation to the DEM analysis part? Scientifically using SWE maps are for validation.

We used the two best DEMs, one from local and one from global sources, as inputs to a 1D-2D flood model to simulate flood inundation. The simulated flood map was then evaluated against a satellite-derived Surface Water Extent (SWE) map. We will refine the abstract to make it more precise.

**We have revised the abstract on lines 1 – 31.**

3. The same applies to the title. Please use a more precise title (laser altimetry was solely used for performance assessment, same applies for the water extent maps (validation, not for an software/model udgrade itself, …) Something like Influence of DEM quality /Performance assessment using global DEM / …

As explained in response to General comments, we have revised the title.

**We have revised the title and the abstract on lines 1 – 31.**

4. Abstract/Intro: The first part "DEM analysis" evaluates 10 DEMs compared to ICESat-2. Please explain your motivation. -> advantage of EO DEMs. The DEM choise is rather heterogenious -> Please categorize the used 10 DEMs e.g. from satellite to airborne DEMs.

We have revised and motivated advantage of EO DEMs into the introduction section.

**We have revised the section 1: introduction on lines 44 -59.**

5. In General: There might exist some specific/logical requirements for DEMs to test or mentioning in advance if they are suited for flood modeling (e. g. like in Gesch, Front. Earth Sci., 2018, Best Practices for Elevation-Based Assessments of Sea-Level Rise and Coastal Flooding Exposure, https://doi.org/10.3389/feart.2018.00230). Against this background, please justify why you start analyzing for so many DEMs the quality from scratch!

The DEMs selected for this study were chosen due to their widespread use in large-scale flood modeling, availability in both free versions and locally collected surveys from Thai sources, and their variations in resolution, methodologies, and data sources. These DEMs are commonly utilized in global and regional flood studies. By comparing them with ICESat-2 ATL08 data, which serves as a high-accuracy reference, the objective was to evaluate their suitability for flood modeling in the Chao Phraya River basin. This thorough analysis ensures the chosen DEM meets the precision required for large-scale flood simulations.

**We have revised and motivation in the section 1: introduction on lines 44 – 67.**

6. Methodology: Chapter 4, Your goal is to find out the best DEM : better in terms of the most accurate DEM compared to ground control points

The primary objective of this chapter is to identify the most accurate Digital Elevation Model (DEM) by evaluating various DEM products against ground control points. In this analysis, ICESat-2 ATL08 elevation data serves as a high-precision reference, effectively functioning as a "ground truth." This role of ICESat-2 ATL08 should be further articulated to underscore its significance in the validation process. Once the optimal DEM is determined through comparative analysis, it is integrated into a 1D-2D flood model. The resulting simulated

flood inundation map is subsequently validated against satellite-derived Surface Water Extent (SWE) maps to rigorously assess its accuracy in representing real-world flood events.

**We have revised on lines 318-322.**

7. 4.1.1 Not described: how do you use/prepare ICESat-2 ATL03 data? If you don't explain it, omit it.

ICESat-2 ATL03 data represents the raw data collected by ICESat-2 before being processed into ATL08. In this study, we did not use ICESat-2 ATL03 for comparison with DEM products. Therefore, we will exclude the ICESat-2 ATL03 section and provide a more detailed explanation in the ICESat-2 ATL08 section, which is derived from the processed ATL03 data.

**We have revised in section 3.4.1 ATL08 on lines 263-265**

8. Comment: Apart of the point comparison I like the additional value of grid and track-wise comparison.

Thank you for comment. We comparison in three comparisons, we can see the overall comparison from point and see the spatial map from grid comparison and elevation profile from track-wise comparison.

9. Results: Sec. 5.2.1 To what table do you refer with the statement given in l.444 (global DEM products was +1.62 m)? Same for statistics in line 449 (no common global DEM statistic in table given).

We have revised the table 6 as shown below:

Table 6: Table of statistical metrics, comparing 10 DEM products against the ICESat-2 benchmark. The resulting averages are computed across the datasets in study area.

| DEM product | Scale | Statistical method | | | | |
|---|---|---|---|---|---|---|
| | | ME (m.) | MAE (m.) | MSE (m.) | RMSE (m.) | PBIAS (%) |
| LDD | Local | -1.30 | 1.64 | 5.45 | 2.33 | -34.76 |
| JICA | Local | -0.65 | 1.04 | 3.51 | 1.87 | -17 |
| merged LDD-JICA | Local | -0.68 | 1.08 | 3.74 | 1.93 | -15.38 |
| **Average local DEMs** | | **-0.88** | **1.25** | **4.23** | **2.04** | **-22.38** |
| ASTER | Global | +4.77 | 5.57 | 44.28 | 6.65 | 47.71 |
| SRTM | Global | +2.04 | 2.58 | 12.92 | 3.59 | 27.99 |
| MERIT | Global | +1.56 | 1.79 | 6.76 | 2.6 | 22.99 |
| GlO30 | Global | +0.84 | 1.3 | 5.89 | 2.43 | 13.87 |
| FABDEMv1-2 | Global | +0.25 | 0.8 | 3.79 | 1.95 | 4.59 |
| TanDEM-X | Global | +0.94 | 1.73 | 13.29 | 3.65 | 15.24 |
| TanDEM-EDEM | Global | +0.91 | 1.43 | 7.74 | 2.78 | 14.84 |
| **Average global DEMs** | | **+1.62** | **2.17** | **13.52** | **3.38** | **21.03** |

**We have revised the Table 6: Table of statistical metrics, comparing 10 DEM products against the ICESat-2 benchmark. The resulting averages are computed across the datasets in study area on line 492 - 494.**

10. please omit the wording "over- and underestimation" when describing small biases of 1-2 m without regarding the RMSE. (i.e. having an RMSE of 2m: an ME of 1 m is within the noise level! No real over- or underestimation) Please scan the whole document! Better use neutral terms like small positive/negative bias.

Yes, we agree. We have revised all wording "over and underestimation" to positive and Negative bias.

**We have revised on lines 471-472, 483-485, 488, 508, 527, 657-660, and 671.**

11. Please re-work the text in Section 5.3. to make it more readable and comprehensive.

**We have revised on lines 541 – 645.**

12. Section 6.2.: Message is unclear, as the different maps for validation seems to have its deficits.

We have revised Section 6.2 to improve clarity and provide a more detailed explanation of the observed Surface Water Extent (SWE) data from the GISTDA and WorldWater projects. The SWE data were generated using different algorithms and satellite sources, leading to different in the results. These observed datasets were then compared to simulated flood maps derived from various Digital Elevation Models (DEMs).

**We have revised Section 6.2 on lines 679 – 680 and 689-693.**

13. Please just list the used data sets / days in Appendix Tables A1 or omit table completely, Table A2 can be omitted completely. It is a service with regular, almost daily acquisitions.

We have revised Tables A1 and A2 by excluding them and referencing the dataset from the WorldWater project or Sentinel data instead.

**We have excluded the Tables A1 and A2 and revised on lines 285, 546 .**

14. The visualization of the geoid models Fig A2 should be omitted.

**We have excluded it.**

Minor comments:

1. Abstract:" Given the current uncertainties stemming from changes in weather  patterns affecting flooding, reducing inaccuracies in flood models is imperative": Please be more precise. Are the uncertainties improving?

**We have revised on lines 11 – 12.**

2. Include in abstract: Which DEMs are finally used for your flood map analysis and why (motivation to chose one global and one local DEM)

**We have revised on lines 16 -23.**

3. l23-l25: Abstract: Think about your message! EO data for validation of DEM and Flood model maps were used. Conclusion/Result?

**We have revised on lines 16-31.**

4. Line 480 ... which can be attributed to the fact that vegetation and buildings are eliminated in this DEM ...

Yes, the FABDEMv1-2 DEM employs machine learning techniques to effectively remove buildings and forested areas from the elevation data.

**We have revised on lines 500-502.**

5. Line 514: we implemented: please re-word to e.g. "performed test with two DEMs,..."

**We have revised on lines 542.**

6. Figure caption: difference of Fig. caption A7 and A9 not clear; RMSE?

We have revised the caption figure A7 to A9

[Figure]

Figure A 7: The histogram distribution of the mean error (ME), comparing 10 DEM products against the ICESat-2 ATL07 benchmark.

[Figure]

Figure A 8: The histogram distribution of the mean absolute error (MAE), comparing 10 DEM products against the ICESat-2 ATL07 benchmark.

[Figure]

Figure A 9: The histogram distribution of the mean square error (MSE), comparing 10 DEM products against the ICESat-2 ATL07 benchmark.

**We have revised the caption and number of figure A6 to A8 on lines 775-780.**

**Reference**

Frias, M.C., Liu, S., Mo, X., Druce, D., Yamazaki, D., Folkmann, A., Nielsen, K., Bauer-gottwein, P., 2024. Improving 2D hydraulic modeling in floodplain areas with ICESat-2 data : A case study in Upstream Yellow River. EGU General Assembly 2024, Vienna, Austria, 14–19 Apr 2024, EGU24-14669, pp. 24–25. doi:https://doi.org/10.5194/egusphere-egu24-14669

Nandam, V., Patel, P.L., 2024. A framework to assess suitability of global digital elevation models for hydrodynamic modelling in data scarce regions. J. Hydrol. 630, 130654. doi:10.1016/j.jhydrol.2024.130654

---

## Author Response (AR2)

We sincerely thank the reviewers for their valuable comments and suggestions, which have significantly contributed to improving the manuscript. Below, we have reproduced the reviewers' comments in black font, followed by our responses in **blue font**.

**Reviewer 1**

Thank you for incorporating the review remarks. Especially for changing the abstract and the title. It is much clearer now.

Some minor/technical points
Abstract: ICESat: there are now two sentences with similar content, p. 1 l. 19-21. Please unify these.

We agree and have revised it on line 19 to 20.

p.13 l. 272: not valid for ATL03 any more: in "... this study are ATL03 and ATL08" used. A separate table like Table 3 is therefore also unnecessary.

Table 3 summarizes the ICESat-2 products, including data collection periods and reference datums, and therefore should not be removed.

p. 15 l. 305: depicted in ->accessible at

We agree and have revised it on line 283.

Revise of Sec. 5.3:
Section 5.3 is still very technical/repetitive and therefore difficult to understand in its message. Exemplarily:
p.38 l.656f: However, the Model flood map effectively detects the GISTDA flood map as well. -> choice of word: The meaning of "detect" is not clear to me. Do you mean that the model map "represents well" the GISTDA flood map ? (please scan Sec. 5.2 by this wording.)

We agree and have revised from detect to corresponds well in Section 5.3

p. 38 l.657f. : not clear how you can conclude this from the measures "that the Model flood map exhibits improved accuracy in comparison to the GISTDA flood map." -> Do you have a better flood map or other experiences?

In this research, we used high-resolution flood maps from the WorldWater project (up to 10 m), as well as multi-source satellite-derived flood maps from GISTDA. These flood maps

provide greater precision. The flood model should be updated and recalibrated because it was last calibrated using data from 2013–2014, which may not reflect current land use changes. Additionally, Thailand has recently implemented new canal diversion projects that affect flood dynamics.

Figures should be in portrait format. Please turn all Figures in the manuscript into this format. It was fine.

Yes, all figures are in portrait format, but some images use a horizontal format to make them larger and clearer.

**Reviewer 2**

General Comments
This paper, entitled "Enhancing the Performance of 1D-2D Flood Models Using Satellite Laser Altimetry and Multi-Mission Surface Water Extent Maps from Earth Observation (EO) Data" by Theerapol Charoensuk, discusses how to improve 1D-2D flood modeling accuracy by validating DEMs with ICESat-2 data and comparing model-predicted inundations against satellite-derived flood extent maps. The paper has clear methodological steps and demonstrates practical value by comparing modelled flood extents with satellite-derived inundation data. The attempt to incorporate ICESat-2 for DEM validation and to benchmark flood maps is timely and relevant. The manuscript would benefit from further discussion on the surprising result that a globally derived FABDEM (with coarser nominal resolution) seems to outperform the locally acquired LiDAR DEM product in certain respects.

In addition, the choice of the acronym "SWE" for surface water extent may cause confusion with the more commonly known usage, "snow water equivalent." With a more detailed discussion of how and why the global FABDEM can outperform an apparently high-resolution local DEM, plus a reconsideration of the acronym used for flood inundation layers, the manuscript will be substantially clearer. The two major issues are as follows.

(1) Discussion of Why a Global DEM (FABDEM) Outperforms a More Detailed Local DEM
A core finding in your study is that the coarser-resolution FABDEM outperforms—or at least equals—the merged LDD-JICA DEM (derived in part from airborne LiDAR) for certain evaluation metrics. This is somewhat counterintuitive, given that airborne LiDAR typically achieves very high resolution and accuracy. Readers will want a deeper explanation as to why this may have occurred. Potential factors include:
• ICESat-2 Footprint and Sampling: If the ICESat-2 ATL08 data used for validation has footprints or sampling intervals that favor broader-scale features, a coarser DEM may align

better with the reference data.
• Local DEM Data Quality: Airborne LiDAR can theoretically be very precise, but if data collection or processing was suboptimal, it could introduce errors. Factors like incomplete vegetation filtering, outdated surveys, or vertical datum mismatches might degrade its performance.

It would be beneficial to elaborate on each of these points to clarify whether you attribute FABDEM's advantage mainly to differences in how building and vegetation heights were removed, to possible systematic bias in the local DEM, or to the nature of the ICESat-2 validation approach itself. Additionally, highlight any limitations in your current processing steps that may have contributed to this unexpected outcome.

We agree and have revised it in Section 6.1 line 650 - 665

(2) "SWE" Acronym for Surface Water Extent
In many hydrologic and cryosphere studies, SWE commonly stands for "Snow Water Equivalent." Using the same acronym to denote "Surface Water Extent" could confuse readers. It might be preferable to adopt an alternative abbreviation such as IE (Inundation Extent), WSE (Water Surface Extent), or another unambiguous term.

Although you clearly define "SWE" in the text as surface water extent, using a more standard term for inundation areas would improve clarity. Please consider replacing the acronym throughout the manuscript and figures with a term that would not conflict with "snow water equivalent."

Thank you for your suggestion. We understand the potential for confusion between the acronym "SWE" (Surface Water Extent) and "Snow Water Equivalent." However, in this manuscript, SWE is consistently defined as Surface Water Extent, and it is also clearly distinguished from WSE (Water Surface Elevation). As this definition is consistently used throughout the manuscript and figures, we prefer to retain the term "SWE" for clarity within the context of our research.